



# Millennial variations of atmospheric CO₂ during the early Holocene (11.7–7.4 ka)

Jinhwa Shin[1,a], Jinho Ahn[1], Jai Chowdhry Beeman[2], Hun-Gyu Lee[1], and Edward J. Brook[3]

[1]School of Earth and Environmental Sciences, Seoul National University, Seoul, 151-742, Republic of Korea
[2]Laboratoire des Sciences du Climat et de l'Environnement, LSCE/IPSL, CEA-CNRS-UVSQ, Université Paris-Saclay, 91191, Gif-sur-Yvette, France
[3]College of Earth, Oceanic, and Atmospheric Sciences, Oregon State University, Corvallis, OR 97331-5506, U.S.A.
[a]current address: Department of Earth and Atmospheric Sciences, University of Alberta, Edmonton, AB, T6G 2E3, Canada

*Correspondence to*: Jinho Ahn (jinhoahn@snu.ac.kr)

**Abstract.** We present a new high-resolution record of atmospheric $CO_2$ from the Siple Dome ice core, Antarctica over the early Holocene (11.7–7.4 ka) that quantifies natural $CO_2$ variability on millennial timescales under interglacial climate conditions. Atmospheric $CO_2$ decreased by ~10 ppm between 11.3 and 7.3 ka. The decrease was punctuated by local minima at 11.1, 10.1, 9.1 and 8.3 ka with amplitude of 2–6 ppm. These variations correlate with proxies for solar forcing and local climate in the South East Atlantic polar front, East Equatorial Pacific and North Atlantic. These relationships suggest that weak
solar forcing changes might have impacted $CO_2$ by changing $CO_2$ outgassing from the Southern Ocean and the East Equatorial Pacific and terrestrial carbon storage in the Northern Hemisphere over the early Holocene.

## 1 Introduction

Future climate and ecosystem changes due to the continual increase of atmospheric carbon dioxide concentrations caused by human activities are inevitable (IPCC, 2013). Understanding the links between the carbon cycle and climate become important
for accurate projection of future climate change. Atmospheric $CO_2$ is controlled by carbon exchange with ocean and land reservoirs, and increased $CO_2$ in the future and consequent changes in the earth system will in turn impact $CO_2$ levels via feedbacks (Friedlingstein et al., 2006). Due to the limited duration of direct measurements of atmospheric $CO_2$, which only started in 1957 (Keeling, 1960), our understanding of the carbon cycle dynamics is limited on longer time scales. Air bubbles occluded in Antarctic ice cores allow us to reconstruct ancient air and may help us better understand the mechanisms that
control atmospheric $CO_2$ (Ahn and Brook, 2008, 2014; Bereiter et al., 2012; Higgins et al., 2015; Lüthi et al., 2008; Marcott et al., 2014; Nehrbass-Ahles et al., 2020; Petit et al., 1999).

Understanding the carbon cycle during interglacial periods is particularly useful because climate boundary conditions are similar to those of the near future. Previous work on late Holocene $CO_2$ records shows centennial $CO_2$ variability linked with climate, but the control mechanisms remain unclear, in part due to the potential mixture of natural and anthropogenic sources
and sinks (Ahn et al., 2012; Bauska et al., 2015; Etheridge et al., 1996; Goosse, 2010; Indermühle et al., 1999; Rubino et al.,





2013; Ruddiman, 2003, 2007). By contrast, $CO_2$ records for the early Holocene (11.7 to 7.3 ka) should reflect only natural $CO_2$ variability due to a smaller human population (Ruddiman, 2003).

The early Holocene (11.7–7.0 ka), is known as a relatively stable period in comparison with glacial periods. Several authors have linked centennial to millennial variability in the early Holocene to changes in solar forcing, including studies of the

Eastern Equatorial Pacific (Marchitto et al., 2010), North Atlantic (Bond et al., 2001) and the Southern Ocean (Nielsen et al., 2004) with responses in proxy records at ~11.1, 10.1, 9.1 and 8.3 ka linked to solar variability (Bond et al., 2001; Marchitto et al., 2010). A weaker (stronger) solar activity has been linked with increased (decreased) ice-rafted debris in North Atlantic (Bond cycle), dominant El-Nino-like conditions (La Niña–like conditions) in the Eastern Equatorial Pacific, weaker (stronger) Asian monsoons, expansion (reduction) of sea ice in the Southern Ocean and colder (warmer) sea surface temperature in the

Southern Ocean (Bond et al., 2001; Marchitto et al., 2010; Nielsen et al., 2004; Reimer et al., 2004; Vonmoos et al., 2006). However, it is not clear what mechanisms are involved (Bond et al., 2001; Darby et al., 2012; Marchitto et al., 2010).

Atmospheric $CO_2$ on millennial time scales is mainly controlled by exchange with oceanic reservoirs and terrestrial carbon stocks. Existing atmospheric $CO_2$ records from EPICA Dome C (Dome C) show little variability of atmospheric $CO_2$ on millennial time scales from 10.9 to 7.3 ka (Monnin et al., 2001; Monnin et al., 2004). However, high-frequency signals might

be muted due to gas trapping processes at this low-accumulation site (Spahni et al., 2003).

In this study, we measured atmospheric $CO_2$ with ages between 11.7 and 9.0 ka from the Siple Dome ice core. This new record complements the existing Siple Dome $CO_2$ record for 9.0–7.3 ka (Ahn et al., 2014b). With this record, we investigate the relationship between atmospheric $CO_2$ and climate variations on centennial and millennial time scales. Siple Dome benefits from an accumulation rate 4.2 times higher than at EDC and 1.8 times higher than at Taylor Dome (Table 1). A conservative

estimate for the width of the gas age distribution in the Siple Dome record gives ~42 years for the early Holocene (Ahn et al., 2014b). Thus, the Siple Dome ice core allows high temporal resolution and higher quality gas data with a more precise age scale and signals that are much less muted by the gas trapping process. The temporal resolution on average during the early Holocene reaches ~30 yr as compared to ~80 yr in the EDC record.

## 2 Methods

### 2.1 $CO_2$ measurements

247 individual ice samples from 99 depth intervals were measured by needle cracker dry extraction and gas chromatography methods at Seoul National University (SNU) (see Figure S1 in SI (Supplementary Information)). We adopted the well-established measurement methods from Oregon State University (OSU) (Ahn et al., 2009) with minor modifications including sharpening of the tips of ice-crushing pins to increase the gas extraction efficiency, and use of a newer model Agilent 7890

60 gas chromatograph (GC).

Briefly, ice samples were cut and trimmed carefully with a band saw in a –21˚C walk-in freezer at SNU. All visible cracks were removed to eliminate potential $CO_2$ alteration by trapping modern air. An ice sample of ~8–10 g was placed in a double



walled vacuum chamber maintained at about −35˚C using cold ethanol circulation between the walls of the chamber while flowing ultra-pure of $N_2$ gas (99.9999%) into the chamber. The ice sample was crushed in the cooled chamber by 91 steel needles moving straight up and down using a linear motion (bellows) vacuum feedthrough. The liberated air from the ice was collected for 3 min in a sample tube in a cryogenic system maintained at 11 K. The $CO_2$ mixing ratio was determined by the Agilent 7890A GC equipped with a flame-ionization detector, using a Ni catalyst which converts $CO_2$ to $CH_4$ before measurement. Sample air was injected into a stainless steel sample loop and the extracted air from each ice sample was analysed twice. The GC system was calibrated daily with a standard air tank (293.25 ppm $CO_2$, WMOX2007 mole fraction scale, calibrated by US National Oceanic and Atmospheric Administration, Global Monitoring Division). To examine the linearity of the GC, ice samples from five different depth intervals ($CO_2$ concentrations of 239-251 ppm) were analysed with two different air standards (188 and 293 ppm $CO_2$, respectively). The average difference in the results using the different standards was $0.4 \pm 0.9$ ppm ($1\sigma$) (Table S1).

## 2.2 Age scale of the Siple Dome ice core records

The Siple Dome samples are placed on the improved Siple Dome chronology developed by Yang et al. (2017), which is aligned with the Greenland Ice Core Chronology, 2005 (GICC05) using the synchronization of $CH_4$ and $\delta^{18}O_{atm}$ time series. Abrupt $CH_4$ changes have been shown to be synchronous within about 50 years with abrupt climate changes in Greenland during the last glacial period (Baumgartner et al., 2014; Rosen et al., 2014). Using this principle, abrupt changes in the composite Siple Dome $CH_4$ data were aligned with abrupt changes in $\delta^{18}O_{ice}$ from the NGRIP ice core (North Greenland Ice Core Project members, 2004; Rasmussen et al., 2006) at the 8.2 ka event and end of the Younger Dryas (Yang et al., 2017). For the time period of 11.64−8.10 ka, ages were updated from the original chronology of Severinghaus et al. (2009) by interpolating the age offsets at the tie points (Yang et al., 2017). For the time intervals outside of 11.64−8.10 ka, the age difference was set constant with the difference at the closest tie point. The modified gas ages are younger than the Severinghaus et al. (2009) ages by less than ~110 years.

## 2.3 Correlation coefficients between CO₂ and climate proxies

In order to assess the relationship between $CO_2$ and climate, we calculate correlations and estimate leads and lags between $CO_2$ and proxies of solar activity as well as climate proxies thought to be themselves related to solar activity. The Pearson correlation coefficient r is commonly used to verify relationships between variables. However, r does not take chronological uncertainty into account. As such, we apply a Monte Carlo procedure to estimate the correlations between $CO_2$ and climate proxies. In the procedure, we adjust the chronologies of the two series, within their chronological uncertainties, and re-calculate r. We do so 1,000 times for each pair, allowing us to calculate a mean correlation coefficient that is more representative of the relationship between time-uncertain series.



We use a similar method to calculate the significance of this correlation against a random red-noise process. At each of the 1,000 steps, we use an AR(1) model (lag-1 auto regression) to fit the series. We then use these AR(1) characteristics to
randomly generate two synthetic series with the same red noise. Then, we calculate the percentage of correlations between the randomized synthetic series that are lower than the correlation coefficients of the real series to assess the significance of the correlation.

Finally, we can calculate the maximum-correlation lag between the two series. At each step in the 1,000 iteration Monte Carlo procedure, we calculate the lag which gives the maximum correlation by shifting one of the series by 10-year increments, for
constant lags between -200 (a $CO_2$ lead) and 200 years (a $CO_2$ lag). Then, we make a histogram of the calculated maximum-correlation lags, from which the mode can be selected as an approximation of the phasing between the two series. We also report the maximum correlation between the two series, but note that this is not representative of the actual relationship (it simply gives an idea of how much the correlation can be improved by adding a lag between the two series).

## 3 Results

### 3.1 The new high-resolution $CO_2$ record during the early Holocene

We obtained 99 data points that cover 622.14–539.06 m at SNU, corresponding to 11.7–9.0 ka (Figure 1). To extend the record to 7.4 ka, we made a composite dataset using a previous $CO_2$ record from the Siple Dome ice core covering 9.0–7.4 ka measured by the needle cracker system at OSU (Ahn et al., 2014b) (Figure 1). Between 2 and 6 replicates (2.6 and 2.4 on average for SNU and OSU data, respectively) from individual depth intervals were analysed. The standard error of the mean of replicates from the same depth interval was 0.8 and 0.5 ppm on average for SNU and OSU data, respectively. The sampling resolution
is ~30 years for 11.7–9.0 ka and ~15 years for 9.0–7.3 ka.

To make a composite record of atmospheric $CO_2$, we tested for bias between the two data sets. Siple Dome samples from 7 depth intervals between 538.55–490. 16 samples were analysed at both laboratories (Ahn et al., 2014b). The SNU measurements were higher than the OSU measurements by 0.3±0.7 ppm (1σ) on average, indicating that the SNU and OSU
results agree well (Table S2 in SI). The small offset of 0.3 ppm was added to OSU data before combining them with the SNU results.

### 3.2 Comparison with existing $CO_2$ records for the early Holocene

The new atmospheric $CO_2$ record from Siple Dome was compared to the existing $CO_2$ data from Dome C measured using the needle cracker at University of Bern (UB) (Monnin et al., 2001; Monnin et al., 2004) and the existing $CO_2$ data from the WAIS
Divide ice core measured by the needle cracker at OSU (Marcott et al., 2014) (Figure S2 in SI). On multi-millennial time scales, the baseline levels of the Siple Dome and WAIS Divide $CO_2$ records (Marcott et al., 2014) are higher than those from Dome C (Flückiger et al., 2002; Monnin et al., 2004) and Taylor Dome (Indermühle et al., 1999) records (Figure S2 in SI). The $CO_2$ offset between the Dome C and Siple Dome ice cores is 3-6 ppm.



The offset between Siple Dome $CO_2$ data in this study and other $CO_2$ data sets might be related to differences in the analytical

methods used to make the measurements. To examine the inter-laboratory analytical offset, several Taylor Dome ice samples were analysed at OSU (Ahn et al., 2014b). The OSU results were higher than those at UB by 1.5 ppm on average. Taking the analytical offset between OSU and SNU of 0.3±0.7 ppm (1σ) into consideration, the 3-6 ppm $CO_2$ offset between the Siple Dome record and Dome C or Taylor Dome cannot be entirely attributed to experimental offset.

$CO_2$ records can be contaminated by the in-situ production of $CO_2$ caused by carbonate-acid reactions (Anklin et al., 1997;

Barnola et al., 1995; Delmas, 1993; Neftel et al., 1988; Smith et al., 1997a; Smith et al., 1997b). Calcium carbonate ($CaCO_3$) in ice cores can cause carbonate-acid reactions, leading to large scattering of atmospheric $CO_2$ data (Smith et al., 1997b). Antarctic ice cores have relatively low concentrations of carbonates and lower site temperatures compared to Greenlandic ice cores, which reduces the risk of $CO_2$ contamination (Tschumi and Stauffer, 2000). It is estimated that the in-situ production of $CO_2$ for Antarctic ice cores is smaller than 1.5 ppm (Bereiter et al., 2009). In-situ production of $CO_2$ cannot be ruled out but

the effect should not greatly impact the offset between records from the different ice cores.

Oxidation of organic compounds (e.g., $2H_2O_2 + HCHO \rightarrow 3H_2O + CO_2$) can also produce $CO_2$ in ice (Tschumi and Stauffer, 2000). The Dome C site is located further from the ocean than Siple Dome and we therefore expect lower organic content in the Dome C ice. Concentrations of organic compounds at our sampling depths are not available. However, the concentration of oxidant $H_2O_2$ on the top 2.5−100 m in the Siple Dome core is below the detection limit of ~0.02 μM (McConnell, 1997),

although 0.02 μM $H_2O_2$ still has potential to produce $CO_2$ and can increase the mixing ratio in bubbles by 5 ppm given sufficient supply of organic compounds (Ahn et al., 2004).

Nehrbass-Ahles (2017), discussed $CO_2$ offsets during the Holocene (11.7−0 ka). For the early Holocene, five datasets were compared: $CO_2$ from the WAIS Divide ice core measured by the needle cracker at OSU (Marcott et al., 2014), $CO_2$ from the Siple Dome ice core measured by the needle cracker at OSU (Ahn et al., 2014b), $CO_2$ from Law Dome measured by sublimation

at UB (Eggleston, 2015), $CO_2$ from Talos Dome measured by the Centrifugal Ice Microtome (CIM) at UB, and $CO_2$ from Dome C measured by the needle cracker at UB (Monnin et al., 2001; Monnin et al., 2004). The individual datasets show similar patterns of $CO_2$ variability over the early Holocene. Offsets among ice cores are lower during the late Holocene but become larger during the early Holocene. During the early Holocene, the largest offset was between Dome C and Law Dome, up to 12 ppm. On the other hand, the baseline level of the Siple Dome $CO_2$ record is similar to that of the WAIS Divide $CO_2$ and Talos

Dome $CO_2$ records. $CO_2$ levels are lower for ice core sites far away from the Antarctic coast, thus we might hypothesize that the level of $CO_2$ concentration could be related to the in situ conditions (e.g. temperature and accumulation rate and the amount of organic compounds) (Nehrbass-Ahles, 2017). However, the offset among records are relatively small.

To compare the new record to the existing records on millennial time scales, we smoothed and filtered the other records using the same approach applied to the Siple Dome record (Figure 2). The trend of the filtered Siple Dome $CO_2$ change is comparable

with those of Dome C (Monnin et al., 2001; Monnin et al., 2004) and WAIS Divide (Marcott et al., 2014) records, in which sampling resolutions are lower than for Siple Dome (Figure 2). The local $CO_2$ minima at 8.3 and 9.0 ka in the Dome C record appear to predate the corresponding minima in the Siple Dome record by 100-200 years, but the different timing is within the





age uncertainty of both records. The WAIS Divide data are scattered during the early Holocene, but the filtered record shows similar millennial $CO_2$ variations as we see in the Siple Dome records (Figure S2 in SI and Figure 2). The low resolution of

the Taylor Dome record of ~390 years makes it impossible to define millennial variations (Figure S2 in SI).

Carbonate-acid reactions can be related with the millennial time scale $CO_2$ variability, thus we examined the concentration of non-sea-salt Ca (nssCa) ion in the Siple Dome and Dome C ice. The nssCa can be produced in ice by the carbonate-acid reaction or transported as a dissolved form. The nssCa records do not correlate well with the filtered millennial $CO_2$ variations in both Siple Dome (r = −0.33) and Dome C (r = 0.15) records during the early Holocene (Figures S3 and Figure S4 in SI).

Even though the origin of nssCa is not well constrained, we pay attention to the observation that the nssCa trends in Dome C and Siple Dome ice do not agree (Figure S3 and Figure S4 in SI), but millennial $CO_2$ variations do. Thus the millennial $CO_2$ variations are not likely artifacts caused by the carbonate-acid chemical reaction in the Siple ice.

The smaller average amplitude of millennial variations in the Dome C record (1.4 ppm, compared to 3.0 ppm for Siple Dome) can be explained by the lower sampling resolution (~80 years for EDC vs. ~20 years for Siple Dome) and a stronger damping

effect on $CO_2$ concentration change at Dome C due to the slower gas trapping process at Dome C (Spahni et al., 2003). The millennial $CO_2$ variations in the ice cores could be attributed to different degrees of natural chemical reactions in ice, but no available data set supports this possibility. If chemical alteration is the main cause of the millennial-scale $CO_2$ variations, we may expect to observe $CO_2$ age offsets among different cores because of dissimilar ice age-gas age differences. In conclusion, the existing ice core records support the millennial $CO_2$ changes in the Siple Dome record although their temporal resolutions

are not sufficient.

In summary, $CO_2$ data sets from different ice cores share similar trends in $CO_2$ change despite offsets in longer term means of a few ppm. These offsets between the Siple dome $CO_2$ record and others do not impact our conclusions.

### 3.3 Atmospheric CO₂ variations on the millennial time scale during the early Holocene

Figure 1 shows atmospheric $CO_2$ from Siple dome during the early Holocene. Atmospheric $CO_2$ increased by ~8 ppm between

11.7 and 11.3 ka and then decreased by ~10 ppm from 10.9 to 7.3 ka. The rapid $CO_2$ increase at 11.7-11.3 ka might be associated with abrupt warming at the end of the last glacial termination (Marcott et al., 2014; Monnin et al., 2001). The long term $CO_2$ trend is generally similar to that of the major water isotope (δD) variations in Antarctic ice cores reflecting Antarctic temperature variations (Figure S5 in SI).

Using the method detailed in Chowdhry Beeman et al. (2019), we calculated the lag of Siple Dome $CO_2$ at the onset of the

Holocene with respect to Antarctic temperature Stack 3 (ATS3) developed by Buizert et al. (2018) using five records: Dome C, Dome Fuji, Talos Dome, EPICA Dronning Maud Land and WAIS Divide. In the temperature stack, the onset of the Holocene period have occurred at ~11.7 ka (Walker et al., 2009). In the Siple Dome $CO_2$ record, the change is estimated to have occurred at 11.5 ka. We estimate a $CO_2$ lag of 190 yrs, within a range of -89 to 397 yrs (95% probability) at the onset of the Holocene (Figure S6 in SI).



The Siple Dome $CO_2$ record shows multi–centennial to millennial variability of ~2–6 ppm with local minima at 11.1, 10.1, 9.1 and 8.3 ka (Figure 1). These variations resemble variability in other paleoclimate records that has been linked to solar cycle variations on these time scales, as described above (Figure 3). To investigate the relationship between atmospheric $CO_2$ and these previous data sets, the Siple Dome $CO_2$ record and climate proxy records were processed by a 1–yr interpolation, smoothed with a 250–yr running mean, high pass filtered at 1/1800 year$^{-1}$ and resampled every 10 yr, to be able to compare

with the filtered proxy records from Marchitto et al. (2010).

We calculated correlation coefficients between the filtered $CO_2$ and climate proxy series to understand their relationship with atmospheric $CO_2$. All data are on their independent age models. Correlation coefficients, their significance, and maximum correlation lags are shown in Figure 4 and Table 2. The composite $CO_2$ record from the Siple dome is anti-correlated with the stacked IRD record in the North Atlantic (Bond et al., 2001) (r = −0.49± 0.1, $CO_2$ time lag of 120±155 years), SST record in

the eastern equatorial Pacific indicating El Niño–like or La Niña–like conditions (r = −0.41±0.13, $CO_2$ time lag of 50±219 years) (Marchitto et al., 2010), and sea ice in the Southern Ocean (r = −0.35± 0.17, $CO_2$ time lag of 190±228 years) (Nielsen et al., 2004). On the other hand, the $CO_2$ record is correlated with summer sea-surface temperature (SSST) in the Southern Ocean (r = 0.35±0.17, $CO_2$ time lag of 52±228 years) (Nielsen et al., 2004). This results imply a tentative link between atmospheric $CO_2$ variations and climate change on millennial time scales. The time lags might be caused by age uncertainties

of the proxy records and/or response time of atmospheric $CO_2$ to climate change (Bauska et al., 2015; Bereiter et al., 2012; Carvalhais et al., 2014).

Interestingly, the highest anti-correlations we find are between the Siple Dome $CO_2$ record and the $^{14}C$ production rate (r = −0.49±0.12, $CO_2$ time lag of −20±148 years) and $^{10}Be$ flux (r = −0.52±0.08, $CO_2$ time lag of 110±63 years). This suggests that $CO_2$ and solar activity co−vary on millennial time scales (Figure 3 and Table 2). Given these observations, solar activity might

be linked to the atmospheric $CO_2$ variations by the response of carbon cycle to climate change during the early Holocene (11.7–7.0 ka) (Figure 3 and Figure 4).

## 4 Discussion

### 4.1 Possible carbon cycle control mechanisms in the Early Holocene

We observed a close relationship between $CO_2$ and Antarctic temperature Stack 3 (ATS3) developed by Buizert et al. (2018)

on long term scales during the early Holocene (Figure S5 in SI), suggesting that $CO_2$ variations on these time scales might be principally controlled by Southern Ocean processes. Atmospheric $CO_2$ can be controlled by temperature and salinity in the ocean (the solubility pump); solubility of $CO_2$ is greater in cooler and fresh surface waters (Broecker, 2002; Takahashi et al., 1993). The formation of deep water occurs in polar regions with high water density, where surface waters are cold, thus, the oceanic uptake of atmospheric $CO_2$ through this mechanism is stronger in polar regions (Sigman and Boyle, 2000). We

observed a tentative link between atmospheric $CO_2$ and summer sea surface temperature (SSST) from the polar front region of



the South East Atlantic on millennial time scales (Nielsen et al., 2004), which implies that lower SSST in the Southern Ocean may lead to a reduction of atmospheric $CO_2$.

Increased sea ice extent might have blocked release of $CO_2$ from $CO_2$-rich deep water to the atmosphere, and therefore decreased atmospheric $CO_2$ concentration as previously suggested for glacial-interglacial $CO_2$ variations (Stephens and

Keeling, 2000). Our Siple Dome $CO_2$ record is negatively correlated with the sea ice extent in the Southern Ocean, although the sea ice extent reconstruction shown in Figure 2 represents only the east Atlantic region of the Southern Ocean.

Oceanic processes associated with El Niño-like and La Niña-like climate variation could also impact the carbon cycle. Marine sediment cores from the East Equatorial Pacific show that solar activity proxies are well correlated with El Niño-like and La Niña-like climate variations in the East Equatorial Pacific SST proxy record (Marchitto et al., 2010). The East Equatorial

Pacific is the region where $CO_2$-rich deep water upwells. Increased upwelling during La Niña -like conditions and resulting increased $CO_2$ outgassing have been suggested for the $CO_2$ increase during the last deglaciation (Kubota et al., 2014). Siple Dome $CO_2$ is anti-correlated with SST in the East Equatorial Pacific on millennial time scales (Figure 3), which may imply that La Niña-like climate can lead to higher $CO_2$ values.

Terrestrial carbon is involved with photosynthesis and respiration in plants, and with soil respiration (microbial and root

respiration). Thus, terrestrial carbon is mostly controlled by temperature and precipitation (Davidson et al., 2000; Mielnick and Dugas, 2000). On multi−millennial time scales, when temperature in Greenland increases from 10.9 to 7.4 ka, atmospheric $CO_2$ decreases. Expansion of vegetation in the Northern Hemisphere may partially contribute to the decrease in atmospheric $CO_2$ (Indermühle et al., 1999).

A recent high resolution study for the last 1,200 years shows that centennial $CO_2$ variability was mainly controlled by terrestrial

carbon, most likely in the high latitude of the Northern Hemisphere (Bauska et al., 2015). The stacked IRD from the North Atlantic may be used for an indicator of cool conditions in the North Atlantic (Bond et al., 1992; Bond et al., 2001). The strong relationship between IRD and atmospheric $CO_2$ indicates that colder climate in the North Atlantic may lower atmospheric $CO_2$ by impacting terrestrial carbon stocks during the early Holocene.

$\delta^{18}O_{ice}$ from the North Greenland Ice Core Project (NGRIP) ice core (Rasmussen et al., 2006) indicating temperature in

Greenland also reveal millennial local minima at similar time intervals as those of $CO_2$ (~11.4, 10.9, 10.2, 9.3 and 8.2 ka), however, atmospheric $CO_2$ and temperature in Greenland are mismatched at the earliest early Holocene and ~8.2 ka. Thus, there is no significant linear relationship between $CO_2$ and temperature in Greenland on millennial time scales, and our calculation indicates that $CO_2$ leads temperature in Greenland on millennial time scales, though the correlation is still too small to assume any relationship (r = 0.21± 0.07, $CO_2$ time lag of −130±63 years).

Temperature in Greenland during the early Holocene might be partially influenced by the internal climate system or/and by low-latitude solar forcing indirectly. Two main cooling events in Greenland are recorded at ~11.4 and ~8.2 ka (Rasmussen et al., 2007). The well-known 8.2 ka cooling event is mainly influenced by the collapse of the Laurentide ice sheet (Merz et al., 2015) rather than by solar forcing; when temperature was colder in Greenland at ~11.4 ka, solar forcing was higher, not



reaching a minimum at until ~11.2 ka. It is also elusive whether solar forcing has an influence on climate in Greenland at ~11.4
ka (Mekhaldi et al., 2020).

Darby et al. (2012) presented proxy data from the coast of Alaska reflecting the Arctic Oscillation (AO) over the past 9.0 ka.
This study found 1,500-year cycles of AO estimated using a proxy of sea-ice drift in the Arctic Ocean, which is similar to the
cycles of IRD in the North Atlantic. But there is no significant linear relationship between AO and solar activity on millennial
time scales during the 9.0 ka record. Solar activity may not be a direct forcing of the AO during the Holocene (Darby et al.,
2012). AO variations might be driven by the internal climate system or affected by low-latitude solar forcing indirectly. Thus,
it is unclear that Greenland temperature variations may cause atmospheric $CO_2$ variations on millennial time scales during the
early Holocene.

## 5 Conclusion

In this study, we present a 30 yr-resolution $CO_2$ record during the early Holocene. Our data show that millennial atmospheric
$CO_2$ variability of 2−6 ppm correlates with several climate proxies such as IRD in the North Atlantic, sea ice extent in the
Southern Ocean, El Niño–like or La Niña–like conditions in the Eastern Equatorial Pacific, all of which appear to be temporally
linked to solar activity (Bond et al., 2001; Marchitto et al., 2010; Nielsen et al., 2004; Reimer et al., 2004; Vonmoos et al.,
2006). The relationships with the proxies are consistent with changes in several different mechanisms that could impact
atmospheric $CO_2$ on millennial time scales including changing $CO_2$ outgassing from the Southern Ocean and the East
Equatorial Pacific, and changing terrestrial carbon storage in the Northern Hemisphere. Our new observations may improve
our understanding of the relationship between climate and carbon cycle on millennial time scales in the absence of
anthropogenic $CO_2$ perturbations. Further studies should focus on deciphering the $CO_2$ control mechanisms with improved
proxy records and carbon cycle models during interglacial periods.

*Data availability.* All data will be available on PANGAEA (Paleoclimatology database websites).

*Author contributions.* The research was designed by JS, JA and EB. The $CO_2$ measurements were performed by JS with
contributions from HL and JA. The data analyses were led by JS and JCB with contributions from JA. JS wrote the manuscript
with inputs from all authors.

*Competing interests.* The authors declare that they have no conflict of interest.

*Acknowledgements.* Financial support was provided by Basic Science Research Program through the National Research
Foundation of Korea (NRF) (NRF-2015R1A2A2A01003888; NRF-2020M1A5A1110607). This research was also partly
conducted under US NSF grants (OPP 0944764 and ATM 0602395) to EB. Our special thanks go to Eunji Byun, Jisu Choi,





Kyungmin Kim and Jiwoong Yang for analytical assistance and the staff of the National Ice Core Laboratory and Michael Kalk of Oregon State University for ice core curation and processing.



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





**Table 1.** Glaciological characteristics of Antarctic ice cores.

| Core name | Mean Annual Temperature (°c) | Mean Accumulation Rate as Water Equivalent (g cm$^{-2}$yr$^{-1}$ as water equivalent) | References |
|---|---|---|---|
| Siple Dome | -25.4 | 12.4 | Hamilton (2002); Severinghaus et al. (2001); Taylor et al. (2004) |
| Taylor Dome | -42 | 7 | Waddington and Morse (1994) |
| EPICA Dome C | -54 | 3 | Schwander et al.(2001); EPICA Dome C 2001-02 Science and Teams (2002); Tabacco et al.(1998) |
| WAIS  Divide | -31 | 20 | Banta et al.(2008); Morse et al.(2002) |





**Table 2.** Correlation between Siple Dome $CO_2$ record and climate proxy records. Column A shows correlation coefficients between $CO_2$ and proxies with $CO_2$ time lags. Column B shows correlation coefficients between $CO_2$ and proxies without 485 $CO_2$ time lag. "With MC" are mean values from the simulations taking age uncertainties into account. "Without MC" is the classic calculation of correlation, without taking age uncertainty into account. Significance of the lag correlations was assessed against 1,000 repetitions of the lag correlation calculation using synthetic data stochastically generated to have the same red noise characteristics as the original series.

| Proxy records (Reference) | A: Correlation between $CO_2$ and proxies with $CO_2$ time lag (yrs) | | | | B: Correlation between $CO_2$ and proxies without $CO_2$ time lag | |
|---|---|---|---|---|---|---|
| | With MC | | Without MC | | With MC | Without MC |
| | r | Time lag | r | Time lag | r (Significance level) | r |
| $CO_2$ - $^{14}C$ production rate Marchitto et al.(2010); Reimer et al.(2004) | -0.49± 0.12 (0.63) | -20±148 | - 0.76 | 50 | -0.48 (0.98) | -0.70 |
| $CO_2$ - $^{10}Be$ flux from Greenland ice core Finkel and Nishiizumi (1997); Marchitto et al. (2010); Vonmoos et al. (2006) | -0.52± 0.08 (0.64) | 110±63 | - 0.61 | 110 | -0.29 (0.77) | -0.32 |
| $CO_2$ - IRD from the North Atlantic region Bond et al. (2001); Marchitto et al. (2010) | -0.49± 0.1 (0.47) | 120±155 | - 0.73 | 170 | -0.33 (0.84) | -0.21 |
| $CO_2$ - SST from eastern equatorial Pacific Marchitto et al. (2010) | -0.40± 0.13 (0.40) | 50±219 | - 0.61 | 80 | -0.38 (0.95) | -0.55 |
| $CO_2$ - Sea ice in the Southern Ocean Nielsen et al. (2004) | -0.35± 0.17 (0.08) | 190±228 | - 0.57 | 100 | -0.24 (0.76) | -0.48 |
| $CO_2$ - SST in the Southern Ocean Nielsen et al. (2004) | 0.35± 0.17 (0.24) | 52±228 | 0.57 | 30 | 0.35 (0.92) | 0.56 |
| $CO_2$ - NGRIP $\delta^{18}O$ Rasmussen et al. (2006) | 0.21± 0.07 (0.05) | -130±63 | 0.11 | 270 | 0.09 (0.37) | 0.06 |



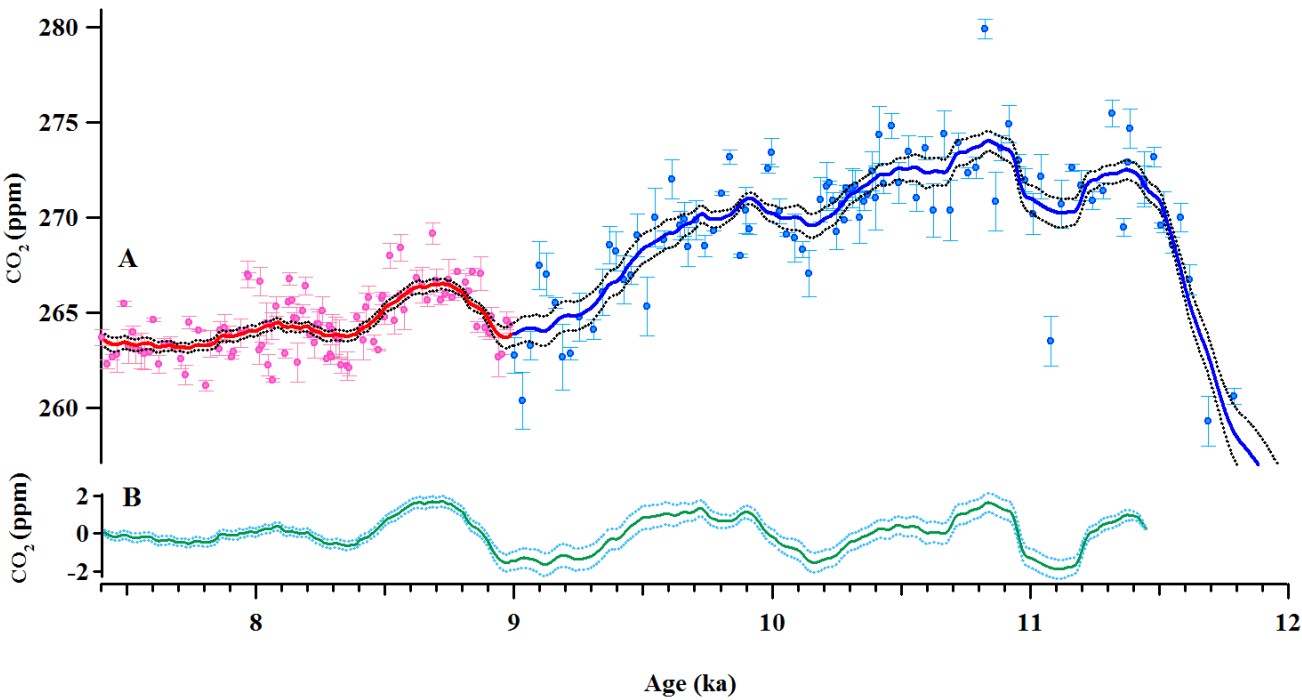


**Figure 1.** High-resolution atmospheric $CO_2$ records obtained from Siple Dome ice core, Antarctica during the early Holocene. A. Pink and blue circles are Siple Dome ice core records obtained at Oregon State University (Ahn et al., 2014b) and Seoul National University (this study), respectively. Lines represent 250–yr running means and dotted lines, $2\sigma$ uncertainties calculated from Monte Carlo simulation. For the simulation, we produced 1000 different sets of $CO_2$ concentrations which
vary randomly with Gaussian propagation in their uncertainties. B. Green line indicates 250−yr running means of the original Siple Dome $CO_2$ data processed by high-pass filtering at 1/1800 year$^{-1}$. Blue line indicates $2\sigma$ uncertainties of calculated from Monte Carlo simulation. The data was detrended by high-pass filtering.



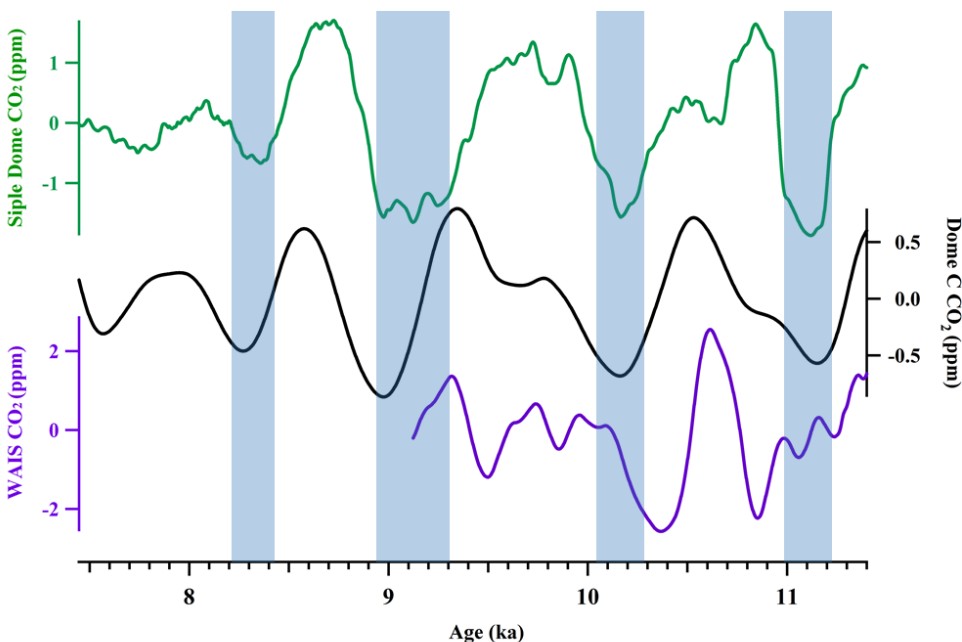

**Figure 2.** Comparison of Antarctic ice core $CO_2$ records. To see long term trend and local minima of these data, the records were smoothed at ~250 years and high-pass filtered at $\frac{1}{1800}$ year$^{-1}$. Green line is Siple Dome ice core records obtained at OSU (Ahn et al., 2014) and SNU (this study). Black line is Dome C obtained from UB (Monnin et al., 2001; Monnin et al., 2004) and purple line indicates WAIS divide ice core from OSU (Marcott et al., 2014).





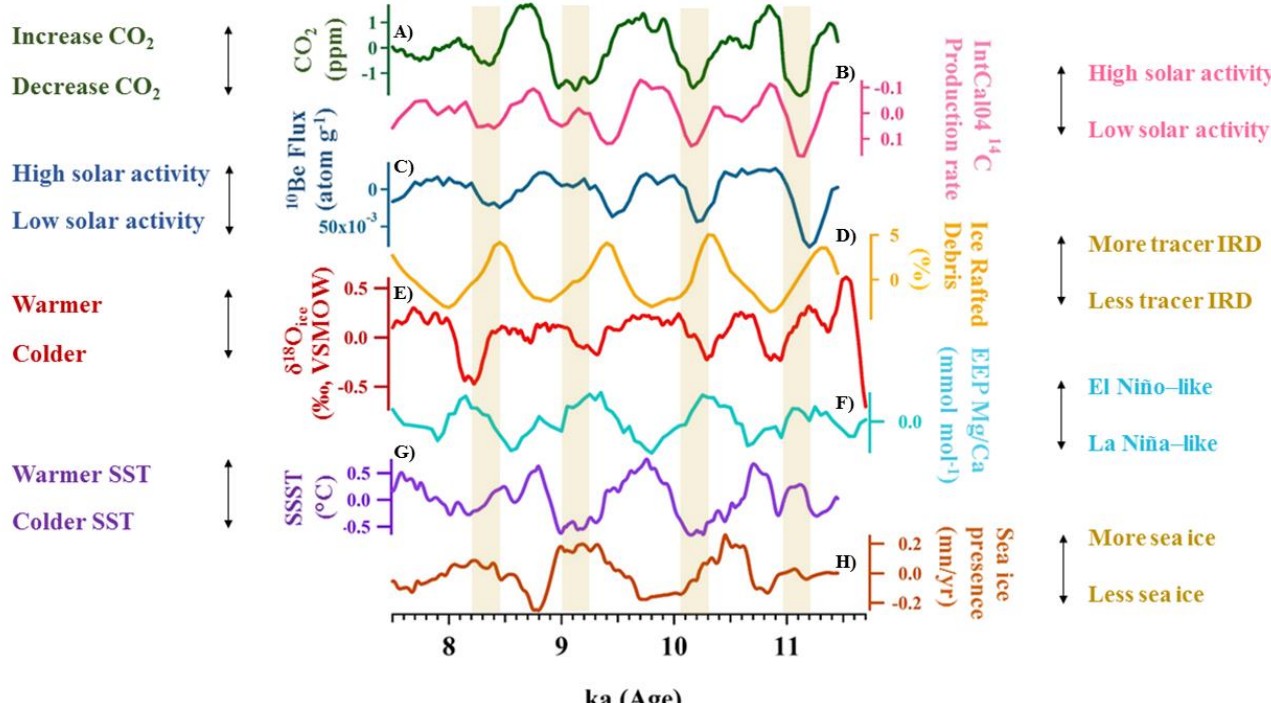


**Figure 3.** Comparison of atmospheric $CO_2$ with climatic proxy records over the early Holocene. The records were smoothed at ~250 years and high-pass filtered at $\frac{1}{1800}$ year$^{-1}$. A) Atmospheric $CO_2$ record from Siple Dome (in this study). B) $^{14}$C production rate from IntCal04 $\Delta^{14}$C data (Marchitto et al., 2010; Reimer et al., 2004). C) $^{10}$Be flux record from ice core on the GICC05 timescale (Finkel and Nishiizumi, 1997; Marchitto et al., 2010; Rasmussen et al., 2006; Vonmoos et al., 2006). D)

IRD stacked records from the North Atlantic regions on untuned calibrated $^{14}$C age model (Bond et al., 2001; Marchitto et al., 2010). E) North Greenland Ice Core Project (NGRIP) ice core isotope ratio on the GICC05 timescale (Rasmussen et al., 2006). F) Sea surface temperature from the eastern equatorial Pacific indicating El Niño–like or La Niña–like conditions (Marchitto et al., 2010). The data was radiocarbon dated by accelerator mass spectrometry (AMS), which was recalibrated by the Marine09 calibration curve (Reimer et al., 2009). G) Sea surface temperature from the Polar Front of the Southern Ocean on the

chronology of Mortyn et al. (2003) (Nielsen et al., 2004). H) Sea ice presence from the Polar Front of the Southern Ocean on the chronology of Mortyn et al. (2003) (Nielsen et al., 2004).





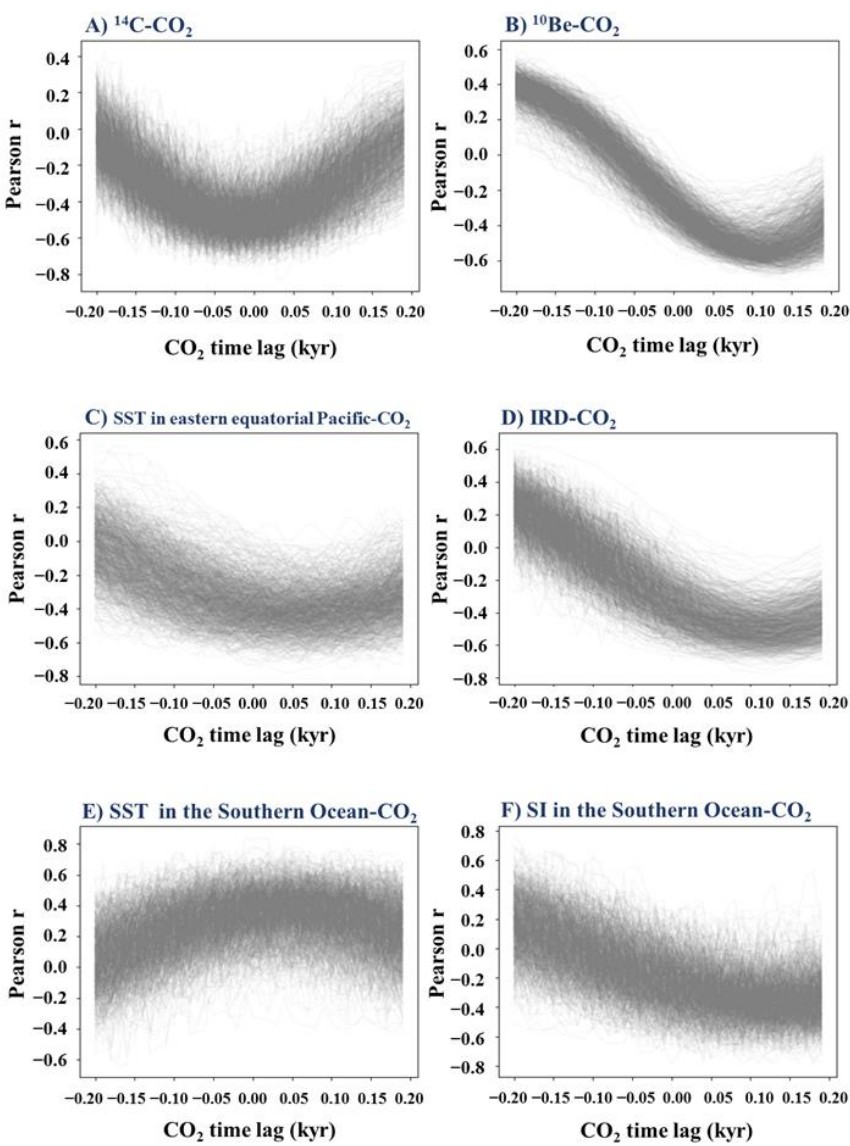

**Figure 4.** Correlation coefficients between $CO_2$ and proxies with $CO_2$ time lag calculated from Monte Carlo simulation. A) $^{14}C$ production rate and atmospheric $CO_2$. B) $^{10}Be$ flux and atmospheric $CO_2$. C) SST in the eastern equatorial Pacific and atmospheric $CO_2$. D) IRD from the North Atlantic and atmospheric $CO_2$. E) SST in the East Equatorial Pacific indicating El Niño–like or La Niña–like conditions and atmospheric $CO_2$. F) SI in the East Equatorial Pacific and atmospheric $CO_2$.