# Peer review of "Millennial variations of atmospheric CO2 during the early Holocene (11.7–7.4 ka)"

_Climate of the Past, 2021_

## Author Response (AR1)

Dear reviewers,

We thank two reviewers for their careful review of our paper, and their suggestions. We appreciate their useful comments and believe their input has improved the paper. Below, we address the comments in blue and the revised texts in the manuscript in green.

All the best, on behalf of all co-authors,
Jinhwa Shin

Referee Comment #1-------------------------------------------------------------------------------------------------------------------

Review of Millennial variations of atmospheric CO2 during the early Holocene (11.7-7.4 ka)

by J. Shin et al.

The authors present a new high-resolution CO2 record, measured on samples from the Siple Dome ice core, from 11.7 to 9 ka and complement an earlier data set from 9 to 7.4 ka from the same core. This results in a high-resolution CO2 record covering the beginning of the Holocene. Interpreting the combined data set they identify small millennial-scale variations of a few ppm and correlate them to various paleoclimate records. The authors speculate that solar irradiation may be responsible for the CO2 variations. The new data, although covering only 2,700 years and thus very short, are important as they close the gap of the early Holocene in the CO2 data of this ice core.

In the present version the authors do not make a sufficiently convincing case for their hypothesis of an influence of solar fluctuations in causing CO2 changes. This is due to (i) questionable data processing that results in very small sigma uncertainties, (ii) essentially correlation-based arguments, (iii) a relatively short discussion of mechanisms, (iv) a nearly inexistent critical reflection on leads and lags that are identified in the data, and (v) a missing credible causal chain from solar fluctuations to purported CO2 variations. Overall, this manuscript requires substantial revisions to reach the maturity of a CP article.

Comments:

1) The interpretation rests on the relatively small CO2 fluctuations that are visible in the filtered data presented in Fig 1. The authors report 2 sigma uncertainties based on Monte Carlo simulations. 2 sigma uncertainties typically contain more than 95% of the data points based on the assumption of normal distribution. The dashed lines in Fig 1, however, are extremely close to the running mean. How can this be? I would have expected a much wider uncertainty band based on the scatter of the data points. Such a wider band would put serious question marks on the significance and robustness of the small fluctuations (few ppm) that are reported in this paper and that are the basis for the claimed sun-CO2 relationship. The authors need to critically revisit the determination and depiction of this 2 sigma uncertainty.

Small variations of atmospheric $CO_2$ are usually smoothed by the gas trapping process in the firn. However, when $CO_2$ data is reconstructed, high-frequency variability of reconstructed $CO_2$ is detected, which might be related to proxy-related noise. Thus, we tried to remove this variability by using a 250-running mean. Due to a 250-running mean, those small variations were removed, thus, the uncertainty band becomes narrow.

The main reason of the small uncertainty is attributed to the removal of the high frequency signal by a 250-running mean as we discussed earlier. When the Monte Carlo simulation was conducted, we considered that each data follows a normal distribution. The width of the error band is affected by neighbouring data points. If the data points are close together, the error of neighbouring data points in the opposite direction can be cancel out, resulting in a narrow uncertainty band.

[Figure]

**Figure 1.** Red circles are Siple Dome ice core records during the early Holocene (11.7-7.4 ka). The black line indicates the average of 1,0000 times modified akima simulations showing an error-weighted average of the $CO_2$ record. The dark shaded indicates $2\sigma$ uncertainties calculated from modified akima simulations.

To assess variability on the millennial time scale, we evaluated 250-year running means and their uncertainties by using a new Monte Carlo approach. Random sampling was made from a probability distribution for each measured value and its standard deviation. If a standard deviation was smaller than the average reproducibility of the measurement ($1\sigma = 0.87$ ppm), we used 0.87 ppm as the uncertainty of a measured value. Then, interpolation

and resampling (1 yr) were applied to generate an evenly-spaced time series and to calculate the 250-year running means. We repeated this series of simulations 10,000 times and evaluated the mean of 250-year running means and its uncertainty (shown as $2\sigma$ in Figure 1). We used a modified Akima method using the built-in makima function in Matlab for the interpolation. The different types of interpolation and smoothing methods resulted in insignificant differences in the 250-year running means.

This is added to supplementary information.

2) The authors use a data processing that is not sufficiently explained. They mention a 1-yr interpolation (line 193) and a 250-yr smoothing, followed by a high-pass filtering at 1/1800 yr-1 and a resampling every 10 years. This sounds like very heavy machinery, and I wonder how robust the results are in light of these interventions. In particular, the 1-year interpolation may add some information to the time series that is simply not inferable from the limited resolution of the measurements and their individual uncertainties. I am very sceptical of this statistical treatment of the data.

250-year running means were made to eliminate high-frequency variability. It is likely that high-frequency variabilities of atmospheric $CO_2$ record (decadal-scale variations and centennial-scale variations) is high frequency noise of atmospheric $CO_2$ record. Thus, we smoothed data sets to eliminate high-frequency variability. Before making a 250-year running mean, we made a 1-year interpolation, because sample spacing between data points covering the early Holocene is not constant. Then, to eliminate this long-term drift of $CO_2$ record, the data was high pass filtered at 1/1800 yr, following previous studies by Bond et al. (2001) and Marchitto et al., (2010). The proxy records were also processed in the same way as the $CO_2$ record to remove high-frequency variability and long-term draft.

It is unlikely that the 1-year interpolation makes any change in our discussion on millennila varitions of $CO_2$ because our original data has a sampling resolution of ~30 years for 11.7–9.0 ka and ~15 years for 9.0–7.3 ka.

This paragraph was revised to: To examine the relationship between atmospheric $CO_2$ and the other paleoproxy data sets on millennial time scales, the Siple Dome $CO_2$ record was smoothed and high pass filtered at 1/1800 yr due to two necessities. First, it is likely that high-frequency variabilities of atmospheric $CO_2$ record (decadal-scale variations and centennial-scale variations) are high frequency noise of atmospheric $CO_2$ record. Thus, we smoothed data sets to eliminate high-frequency variability. Before making a 250-yr running mean, we made a 1-yr interpolation, because sample spacing between data points covering the early Holocene is not constant. Second, to eliminate multi-millennial drift of $CO_2$ record, the data was high pass filtered at 1/1800 yr, following previous methods by Bond et al. (2001) and Marchitto et al., (2010). The proxy records were also processed in the same way as the $CO_2$ record to remove high-frequency variability and long-term draft.

Furthermore, all other paleoclimate data are treated with the same method, and without showing the original data points of these records the authors do not make a convincing case for the significance of such small variations. In short, the data treatment is insufficiently described, and a robustness analysis is missing.

We modified figure 2 so that we may present all the original records as well as 250-year running means (figure 2). The smoothing by a 250-year running is necessary to clearly show the millennial variations. To detrend the long-term change and focus on millennial variations, we can apply the high-pass filtering at 1/1800 year$^{-1}$. Regardless of the high-pass filtering, the 250-year running means show similar millennial variations among the multiple proxy records.

[Figure]

**Figure 2.** Comparison of atmospheric $CO_2$ with climatic proxy records over the early Holocene. Lines represent 250–yr running means. (A) IntCal20 $^{14}C$ production rate (Reimer et al., 2020). (B) Ice rafted debris stacked records from the North Atlantic regions on untuned calibrated $^{14}C$ age model (Bond et al., 2001; Marchitto et al., 2010). (C) Sea surface temperature from the eastern equatorial Pacific indicating El Niño–like or La Niña–like conditions (Marchittoet al., 2010). (D) Sea ice presence from the Polar Front of the Southern Ocean on the chronology of Mortyn et al. (2003) (Nielsen et al., 2004). (E) Sea surface temperature from the Polar Front of the Southern Ocean on the chronology of Mortyn et al. (2003) (Nielsen et al., 2004). (F)  Atmospheric $CO_2$ record from Siple Dome (in this study).

This figure is added to supplementary information.

3)  Panel B of Fig 1 shows the high-pass filtered signal. Peak-to-peak amplitudes are max 4 ppm with some fluctuations of less than 1 ppm relative to the mean. So far, such small variations in CO2 measured from ice core samples, have not been interpretable, given the typical measurement uncertainties that are known from the literature. The authors have the burden of making a convincing case that such fluctuations here can indeed be interpreted as variations in atmospheric CO2 concentrations.

Previous paleoclimate studies showed climate fluctuations on millennial time scales with the local minima at around ~11.1, 10.3, 9.4, 8.1 ka covering the early Holocene (Bond et al., 2001; Marchitto et al., 2010; Nielsen et al., 2004; Reimer et al., 2004; Vonmoos et al., 2006). In this study, we wished to focus on how this millennial climate fluctuations affect atmospheric $CO_2$. The data interpretation was mainly about the millennial-scale variation of atmospheric $CO_2$ with the amplitude of ~4 ppm. These variations are relatively small compared to $CO_2$ variations during other periods such as glacial periods. However, the amplitude was calculated using a smoothing curve, which implies that artificial noise made by the gas trapping process in the firn was deleted. In addition, as we discussed earlier, the types of interpolation and smoothing methods do not resulted in significant differences in the 250-year running means. Thus, these variations by ~4 ppm cannot be ignored.

4) Comments 1 to 3 also apply to Fig. 2. For the CO2 data from EDC (Monnin et al) and WAIS (Marcott et al), the curves are misleading. Inspecting the original data in these papers, I am not convinced that the fluctuations that are shown in the processed data exhibit a robust signal that would represent atmospheric variations. Here a much more careful analysis and statistical assessment (see comment 2) would have to be carried out to see whether such small CO2 fluctuations can be identified in all three ice cores. It appears on the basis of the presented information that the authors go too far in their interpretation for this relatively short record.

[Figure]

**Figure 3.** A: Atmospheric $CO_2$ records. Red dots: Atmospheric $CO_2$ record from Dome C ice core. Red line: 250-running mean of atmospheric $CO_2$ record from Dome C ice core. Blue dots: Atmospheric $CO_2$ record from Siple Dome ice core. Blue line: 250-running mean of atmospheric $CO_2$ record from Siple Dome ice core. Green dots: Atmospheric $CO_2$ record from WAIS Divide ice core. Green line: 250-running mean of atmospheric $CO_2$ record from WAIS Divide ice core. B: $CO_2$ offset between Siple Dome $CO_2$ record and other published $CO_2$ records. Red line: $CO_2$ offset between Siple Dome $CO_2$ record and Dome C $CO_2$ record. Green line: $CO_2$ offset between Siple Dome $CO_2$ record and WAIS divide $CO_2$ record.

To verify the levels of agreement, we calculate the Pearson correlation coefficient between Siple Dome $CO_2$ and existing $CO_2$ records (Figure 3). The offsets between existing $CO_2$ records and our data are also calculated (Figure 3). Before calculating the correlation coefficient between Siple Dome $CO_2$ record and other $CO_2$ records, all $CO_2$ records were smoothed by 250-running means to eliminate high frequency noise.

We observe that $CO_2$ data sets from Siple dome and Dome C share similar trends in $CO_2$ variations despite the $CO_2$ offset in longer term means of 3~8 ppm. The $CO_2$ record from the Siple dome is highly correlated with the $CO_2$ record from Dome C (r= 0.89). The $CO_2$ offset between Dome C record and Siple Dome record decreases continuously from 11.7 ka to 7 ka with small variations at around 9.3 and 8.3 ka (Figure 3B). The small variations of Dome C $CO_2$ record might be caused by the low sampling resolution and a stronger damping effect on $CO_2$ concentration change due to the slower gas trapping process at the Dome C site (Spahni et al., 2003).

The Correlation coefficient between Siple Dome $CO_2$ and WAIS divide $CO_2$ is 0.7. However, the $CO_2$ offset between Dome C record and Siple Dome record is quite random (Figure 3B) because of scattering in the WAIS Divide $CO_2$ record during the early Holocene period.

The WAIS Divide $CO_2$ data was reconstructed from the ice just below the bubble clathrate transition zone (BTCZ). Previous studies raised an issue about the possibility of high frequency noise of atmospheric $CO_2$ record in the ice just below the bubble clathrate transition zone (Lüthi et al., 2010; Shackleton et al, 2019). This phenomenon might be related to gas fractionation effect because of clathrate layering during bubble-clathrate transformation. Gas content starts to be fractionated in the BCTZ because of the differential permeation of gas species when bubbles have transformed to clathrates. $CO_2$ concentration in the first layer of clathrates is more enriched with higher bubble-to-clathrate permeation rates. Below the BCTZ, gas content slowly homogenizes again through molecular diffusion (Bereiter et al., 2009), which can cause high frequency noise to the ice below the BCTZ. Thus, the WAIS Divide $CO_2$ data is not sufficient to discuss millennial variabilities of the early Holocene.

In short, our comparison with the Dome C and WAIS Divide records supports our Siple Dome record although the comparison is limited due to relatively large offsets and low data resolution of the existing records. We will add words in the text to clarify the limits of the assessment.

This is written to Section 3.2 Comparison with existing $CO_2$ records for the early Holocene

Fig 3 suggests evident leads and lags, but they are not discussed and explained in the text. If it turns out that the fluctuations are robust, then these leads and lags need to be considered and discussed in the context of mechanisms. They may be helpful in constraining the causal chain, if such does indeed exist, from solar variations to the CO2 fluctuations. The authors end their discussion at a correlation analysis among the different paleoclimatic records of Fig 3. Correlation is not causation, and thus the arguments for a solar connection to CO2 fluctuations is rather weak, if it is active at all.

Figure 3 shows a possible link between solar activity/climate variations and atmospheric $CO_2$ during the early Holocene. Unfortunately, each data has their own age scale, thus with this figure, it is difficult to discuss a link between solar activity and atmospheric $CO_2$.

We found maximum correlation coefficients between $CO_2$ and climate proxies with $CO_2$ time lags of ~90 years (Table 2). Thus, we assumed that atmospheric $CO_2$ might be affected by solar activities via climate (Figure 4 and Table 2). Changes of solar activities may impact on stratospheric ozone concentrations, which can change stratospheric and tropospheric circulation patterns (Meehl et al., 2009). Higher solar activity enhanced precipitation in the Intertropical Convergence Zone (ITCZ) and South Pacific Convergence Zone (SPCZ) (Meehl et al.,2009; van Loon et al., 2007). The intensified moisture at those areas increased trade wind strength and upwelling in the East Equatorial Pacific region. These conditions lead to Na Niña like climate states on millennial time scale (Marchitto et al.,2010). This ocean condition change in the East Equatorial Pacific has affected the North Atlantic (Darby et al., 2012).

This is written to Section 4.1 Possible carbon cycle control mechanisms in the Early Holocene:

Understanding a link between climate variations and solar activity on millennial time scales during the early Holocene is important to decipher carbon cycle mechanisms. However, the climate mechanisms have not yet been deciphered. A possible mechanism is that changes of solar activities may impact on stratospheric ozone concentrations, which can change stratospheric and tropospheric circulation patterns (Meehl et al., 2009). Higher solar activity may enhance the precipitation in the Intertropical Convergence Zone (ITCZ) and South Pacific Convergence Zone (SPCZ) (Meehl et al.,2009; van Loon et al., 2007). Consequently, the intensified moisture at those areas would increase trade wind strength and upwelling in the East Equatorial Pacific region. These conditions would lead to Na Niña like climate states on millennial time scale (Marchitto et al., 2010). This change in the East Equatorial Pacific might have affected the North Atlantic (Darby et al., 2012).

5) Further to Fig 3, age scale uncertainties between the different records seem to be ignored. These would represent an additional significant uncertainty regarding leads and lags. It is evidently difficult to come up with a common age scale, but the minimum expected would be an assessment the consequences for the conclusions.

Each record has their own age scales. Thus, as the reviewer mentioned, we may miss the relationship between $CO_2$ and proxies due to the age uncertainties of proxies and $CO_2$ records. This would be the case for centennial variations. As we focus on millennial variations, we check the leads and lags which are mostly smaller than 200 years. Even we consider the age uncertainties, we found similar variations of proxies and $CO_2$ on millennial time scales during the early Holocene. Figure 3. Figure 4 and Table 2 show the relationship between $CO_2$ and proxies by considering age uncertainties.

6) ENSO is offered as one of the possible mechanisms for CO2 fluctuations (lines 227ff). The discussion is rather superficial and incomplete. While Feely et al (1999) identify a decrease of CO2 during the 1991-94 El Nino, Chatterjee et al (2017) provide a more detailed, satellite-based analysis of the effects of the 2015-16 El Nino. After an initial decrease, consistent with Feely et al, they observe a stronger increase in the later stages of this El Nino, with the overall result of a CO2 increase. The two El Nino episodes are quite different with the former persisting for 3 years, while the later lasts for only one year but is stronger. Therefore, it seems not robust to assign a clear correlation between small OC2 fluctuations and ENSO.

Thank you for your suggestion. As you suggested, we considered adding this information about two oppositional observations in the manuscript. However, the time scale is one of the important factors to interpret $CO_2$ records. Because depending on the time scale, the role of the ocean also can vary (Gottschalk et al., 2019). Two papers are about the relationship between $CO_2$ and ENSO on the annual time scale. Thus, it is not necessary to apply the annual observations to the relationship between $CO_2$ and ENSO on the millennial time scale.

7) Lines 256ff on a possible role of the AO are speculative and do not add substance to the paper. Either this connection should be explored more in depth or deleted.

Deleted.

**References**

Bond, G., Kromer, B., Beer, J., Muscheler, R., Evans, M. N., Showers, W., Hoffmann, S., Lotti-325 Bond, R., Hajdas, I., and Bonani, G.: Persistent solar influence on North Atlantic climate during the Holocene, Science, 294, 2130–2136, 2001.

Darby, D. A., Ortiz, J., Grosch, C., and Lund, S.: 1,500-year cycle in the Arctic Oscillation identified in Holocene Arctic sea ice drift, Nat. Geosci., 5, 897–900, 2012.

Gottschalk, J., Battaglia, G., Fischer, H., Frölicher, T. L., Jaccard, S. L., Jeltsch-Thömmes, A., Joos, F., Köhler, P., Meissner, K. J., and Menviel, L., Nehrbass-Ahles, C., Schmitt, J., Schmittner, A., Skinner, L. C., and Stocker, T.: Mechanisms of millennialscale atmospheric CO2 change in numerical model simulations, Quaternary Sci. Rev.,220, 30–74, 2019.

Marchitto, T. M., Muscheler, R., Ortiz, J. D., Carriquiry, J. D., and van Geen, A.: Dynamical response of the tropical Pacific Ocean to solar forcing during the early Holocene, Science, 330, 1378–1381, 2010.

Meehl, G. A., Arblaster, J. M., Matthes, K., Sassi, F., and van Loon, H.: Amplifying the Pacific Climate System response to a small 11-year Solar Cycle forcing, Science, 325, 1114–1118, doi:10.1026/science.1172872, 2009.

Nielsen, S. H. H., Koc, N., and Crosta, X.: Holocene climate in the Atlantic sector of the southern ocean: Controlled by insolation or oceanic circulation?, Geology, 32, 317–320, 2004.

Reimer, P. J., Austin, W. E. N., Bard, E., Bayliss, A., Blackwell, P. G., Ramsey, C. B., Butzin, M., Cheng, H., Edwards, R. L., Friedrich, M., Grootes, P. M., Guilderson, T. P., Hajdas, I., Heaton, T. J., Hogg, A. G., Hughen, K. A., Kromer, B., Manning, S. W., Muscheler, R., Palmer, J. G., Pearson, C., van der Plicht, J., Reimer, R. W., Richards, D. A., Scott, E. M., Southon, J. R., Turney, C. S. M., Wacker, L., Adolphi, F., Büntgen, U., Capano, M., Fahrni, S. M., Fogtmann-Schulz, A., Friedrich, R., Köhler, P., Kudsk, S., Miyake, F., Olsen, J., Reinig, F., Sakamoto, M., Sookdeo, A., and Talamo, S.: The IntCal20 Northern Hemisphere radiocarbon age calibration curve (0–55 cal. ka BP), Radiocarbon, 62, 725–757, https://doi.org/10.1017/RDC.2020.41, 2020.

Shackleton, S., Bereiter, B., Baggenstos, D., Bauska, T. K., Brook, E. J., Marcott, S. A., and Severinghaus, J. P.: Is the Noble Gas-Based Rate of Ocean Warming During the Younger Dryas Overestimated?, Geophys. Res. Lett., 46, 5928–5936, https://doi.org/10.1029/2019GL082971, 2019.

van Loon, H., Meehl, G. A., and Shea, D. J., Coupled air-sea response to solar forcing in the Pacific region during northern winter, J. Geophys. Res., 112, D02108, doi:10.1029/2006JD007378, 2007.

Vonmoos, M., Beer, J., and Muscheler, R.: Large variations in Holocene solar activity: Constraints from 10Be in the Greenland Ice Core Project ice core, J. Geophys. Res.-Space, 111, A10105, doi:10.1029/2005JA011500, 2006.

Dear reviewers,

We thank two reviewers for their careful review of our paper, and their suggestions. We appreciate their useful comments and believe their input has improved the paper. Below, we respond to the comments in blue and the revised texts in the manuscript in green.

All the best, on behalf of all co-authors,
Jinhwa Shin

Referee Comment #2--------------------------------------------------------------------------------------------------------

**Summary:**

Shin et al., present a novel, high-quality dataset of atmospheric CO2 during the early Holocene. Using the relatively high-resolution Siple Dome ice core, they provide an atmospheric CO2 record with the best resolution possible in this underexplored time interval. They identify previously un-resolved millennial-scale variations in atmospheric CO2. Relying on statistical inferences, they suggest that solar variations may be the primary forcing of the small changes in atmospheric CO2 (2-6 ppm). Various mechanisms to link the solar variability with atmospheric CO2 level operating in the EEP, Southern Ocean and the Northern Hemisphere terrestrial biosphere are discussed.

Shin et al., provide an excellent new ice core CO2 dataset that will be of wide interest to the palaeoclimate and carbon cycle community and should certainly be published. Crucially, the record extends our knowledge about fine-scale changes in CO2 into the early Holocene. The early Holocene is an important interval to study as it gives us clues about the sensitivity and stability of the carbon cycle during past warm intervals. In the early Holocene, CO2 and global temperature have reached interglacial levels and NH temperature may have risen to peak Holocene levels (Marcott et al., 2013, albeit this is likely limited to marine margins e.g Marisiek et al., 2018; Kaufmann et al., 2020), the large ice sheets of the NH were still in retreat, and the NH terrestrial biosphere was undergoing a substantial amount of regrowth (Elsig et al., 2009).

Overall, I felt the paper makes a though-provoking observation that small, high-frequency changes in CO2 may be correlated to solar variability. However, the causal links explored in the paper are not very convincing. Because the timescales of solar variability are wide-ranging (~decades to millennia) yet highly uncertain, they can easily be misattributed other processes in the climate system (e.g. internal climate variability) or noise in a proxy system. Thus the bar should be set very high for hypotheses invoking solar forcing based purely on statistical methods.

I also had some questions about how the raw data were transformed into robust time series for analysis. Finally, although the paper is quite brief, the organization could use some improvement as I found myself having to jump around looking for key information.

I look forward to reading a revised manuscript on this exciting new dataset.

**1) Testing the solar hypothesis**

In parts of the paper, the authors compare and contrast the variability they observe in the early Holocene to that relatively well-known centennial-scale variability of the pre-Industrial late Holocene. The premise of the paper is in part setup as a test of whether Anthropogenic emissions are the main driver of high-frequency CO2 variability. But this test is never followed up on. The reader is left with some questions:

• Given the proposed link between solar and $CO_2$ in the early Holocene, what is the predicted influence of solar variability in the late Holocene (when solar forcing and climate variability are much better constrained)?
• Do the mechanisms proposed agree with the late-Holocene CO2 data and thus support the author's hypothesis?
• If not, does it none-the-less challenge our understanding CO2 in the late Holocene?   Is there another way to test this hypothesis?
• Is there something different in the carbon cycle boundary conditions that make the comparison between the early and late Holocene difficult?
• Finally, it would worthwhile pointing out that any changes in CO2 due to solar variability are very small and, even if present, would be swamped by changes in anthropogenic emissions during the Industrial Period.

The Holocene period (11.7–0 ka) was previously considered as a stable and warm period. Recent studies revealed climate variations on centennial to millennial timescale covering the Holocene thanks to high-resolution proxy records. Bond et al., (2001) first revealed millennial-timescale variations of ice-rafted debris (IRD) in North Atlantic with local minima at ~11.1, 10.3, 9.4, 8.1, 5.9, 4.2, 2.8, 1.4 and 0.4 ka, which is called the Bond cycle (Bond et al., 2001). Likewise, proxy records from Ocean sediments, tree ring and stalagmite in some regions also show similar variations of climate on millennial time scales (Bond et al., 2001; Liu et al., 2020; Marchitto et al., 2010; Nielsen et al., 2004; Reimer et al., 2004; Varma et al., 2011; Vonmoos et al., 2006). These variations of climate proxies resemble solar variations on millennial time scales, implying that millennial variations of climate were thought to be influenced by variation of solar activity (Bond et al., 2001).

Although millennial variations of climate covering the Holocene are observed, there are similarities and differences between the early Holocene and the late Holocene (Liu et al., 2020). During the early Holocene (11.7–7 ka), the solar activities on millennial time scales are prominent. IRD data and solar forcing show a periodicity at 1.0 kyr. These observations indicate that solar forcing ($^{14}$C) may have an important role in climate change on millennial time scales (Liu et al., 2020).

[Figure]

**Figure 1.** Atmospheric $CO_2$ from Antarctic ice cores during 7-0 ka. Blue dots: atmospheric $CO_2$ from Dome C (Monnin et al., 2001, 2004). Red line: atmospheric $CO_2$ from Law Dome (MacFarling Meure et al., 2006). Yellow dots: atmospheric $CO_2$ from WAIS Divide (Ahn et al., 2012). Green dots: atmospheric $CO_2$ from EDML (Monnin et al., 2004; Siegenthaler et al., 2005). Black dots: atmospheric $CO_2$ from Law Dome and South Pole (Rubino et al., 2013).

During the late Holocene (7–1 ka), the solar forcing on millennial time scales is moderate. The dominant periodicity of IRD and solar forcing proxy is 1.27 kyr and 0.71 kyr respectively. Given these observations, solar forcing during the late Holocene may not play a prominent part in millennial variabilities of climate (Liu et al., 2020). Thus, comparing $CO_2$ variations during the early Holocene and the late Holocene would be great to understand a link between atmospheric $CO_2$ and solar forcing. However due to the lack of high-resolution records of atmospheric $CO_2$ during the late Holocene, observing millennial variations of atmospheric $CO_2$ was limited (Figure 1).

[Figure]

**Figure 2.** Atmospheric $CO_2$ from Antarctic ice cores during the last 2,000 years. Blue line: total solar irradiance (TSI) (Roth and Joos, 2013). Yellow dots: atmospheric $CO_2$ from WAIS Divide ice core (Ahn et al., 2012). Green dots: atmospheric $CO_2$ from EDML (Monnin et al., 2004; Siegenthaler et al., 2005). Black dots: atmospheric $CO_2$ from Law dome (Rubino et al., 2019).

Alternatively, we investigate a possible link between atmospheric $CO_2$ variations and solar activities during the last 1,000 years. Several studies suggested that the variation in total solar irradiance over the 11-year sunspot cycle accounts for less than 0.1% (Lean. 2000), the greater relative variation of solar ultraviolet output caused by sunspot cycle may affect stratospheric ozone concentrations, therefore impacting on the stratospheric and tropospheric circulation patterns (Meehl et al., 2009).

Figure 2 shows Total Solar Irradiance (TSI) and atmospheric $CO_2$ during the last 2,000 yrs. There are two periods in which sunspots were exceedingly rare. During the Maunder sunspot minimum (1647–1715 CE), TSI was reduced by $0.85 \pm 0.16$ W m$^{-2}$. Atmospheric $CO_2$ records from Antarctic ice cores commonly show a decrease trend during this period (Ahn et al., 2012; Monnin et al., 2004; Siegenthaler et al., 2005; Rubino et al., 2019). During the Spörer Minimum (1450–1550 CE), TSI record during this period also shows a decrease trend. However, atmospheric $CO_2$ decrease is not significant in Law and EDML records (Monnin et al., 2004; Siegenthaler et al., 2005; Rubino et al., 2019) while WAIS divide ice shows a decrease during this period (Ahn et al., 2012) (Figure 2). Thus, using $CO_2$ records during the late Holocene, it is limited to understand how atmospheric $CO_2$ is related with solar activity. To improve our understanding of a possible link between atmospheric $CO_2$ variations and climate variations caused by solar activity, improving the resolution of atmospheric $CO_2$ records covering the late Holocene would be helpful.

This is written in the revised manuscript in 4.1 Possible carbon cycle control mechanisms in the Early Holocene:

A positive correlation between solar forcing and atmospheric $CO_2$ is observed during the Little Ice Age (LIA). There are two periods in which sunspots were exceedingly rare. During the Maunder sunspot minimum (1647–1715 CE), total solar irradiance (TSI) was reduced by $0.85\pm0.16$ W m$^{-2}$. Atmospheric $CO_2$ records from Antarctic ice cores commonly show a decrease trend during this period (Ahn et al., 2012; Monnin et al., 2004; Siegenthaler et al., 2005; Rubino et al., 2019). During the Spörer Minimum (1450–1550 CE), TSI record during this period also shows a decrease trend. However, atmospheric $CO_2$ decrease is not significant in Law Dome and EPICA Dronning Maud Land (EDML) records (Monnin et al., 2004; Siegenthaler et al., 2005; Rubino et al., 2019), while WAIS divide ice record shows a decrease during this period (Ahn et al., 2012) (Figure S7 in SI).

2)  **Error propagation in high-pass filtering and subsequent analysis**

It was not clear to me if the analytical precision of the CO2 data was included in the high-pass filtering and also the lag correlations. This could be crucial in demonstrating the robustness of the results. For example, no uncertainty bands are provided in Figures 2 and 3.

Figure 3 is revied (Please see below). Figure 2 is removed from the manuscript.

I could only a find a reference to the uncertainty in the measurements being included in caption for Figure 1. Please provide more detail in the methods section. Also, the exact the method used to obtain the high-pass filtered time series is not mentioned. Similarly to the above, the cut-off frequency for the CO2 curve was only mentioned in the figure captions.

250-year running means were calculated to eliminate high-frequency variability. Small variations of atmospheric $CO_2$ are usually smoothed by the gas trapping process in the firn. It is likely that decadal- to centennial-scale

variabilities of atmospheric $CO_2$ record can be attributed to the noise of the atmospheric $CO_2$ record. Before calculating a 250-year running mean, we made a 1-year interpolation, because sample spacing between data points covering the early Holocene is not constant. Then, to eliminate this long-term drift of $CO_2$ record, the data was high pass filtered at 1/1800 yr, following previous studies by Bond et al. (2001) and Marchitto et al., (2010). We evaluated uncertainties of $CO_2$ record from Siple Dome by using a Monte Carlo simulation. For the simulation, we produced 1000 different sets of $CO_2$ concentrations which vary randomly with Gaussian propagation in their uncertainties.

[Figure]

**Figure 3 in the revised manuscript.** Comparison of atmospheric $CO_2$ with climatic proxy records over the early Holocene. The records were smoothed at ~250 years and high-pass filtered at 1/1800year$^{-1}$. A) Atmospheric $CO_2$ record from Siple Dome (in this study). Dotted lines, 2σ uncertainties calculated from Monte Carlo simulation. B) $^{14}$C production rate from IntCal04 $\Delta^{14}$C data (Marchitto et al., 2010; Reimer et al., 2004). C) $^{10}$Be flux record from ice core on the GICC05 timescale (Finkel and Nishiizumi, 1997; Marchitto et al., 2010; Rasmussen et al., 2006; Vonmoos et al., 2006). D). IRD stacked records from the North Atlantic regions on untuned calibrated $^{14}$C age model (Bond et al., 2001; Marchitto et al.,2010). E) North Greenland Ice Core Project (NGRIP) ice core isotope ratio on the GICC05 timescale (Rasmussen et al., 2006). F) Sea surface temperature from the eastern equatorial Pacific indicating El Niño–like or La Niña–like conditions (Marchitto et al., 2010). The data was radiocarbon dated by accelerator mass spectrometry (AMS), which was recalibrated by the Marine09 calibration curve (Reimer et al., 2009). G) Sea surface temperature from the Polar Front of the Southern Ocean on the chronology of Mortyn et al. (2003) (Nielsen et al., 2004). H) Sea ice presence from the Polar Front of the Southern Ocean on the chronology of Mortyn et al. (2003) (Nielsen et al., 2004).

This is written in the revised manuscript in 3.3 Atmospheric $CO_2$ variations on the millennial time scale during the early Holocene

3) **Comparison to other $CO_2$ records**

The WAIS Divide data in this interval lies just below the bubble clathrate transition zone. As first discussed in Lüthi et al. 2010 and shown specifically for WAIS Divide in Shackleton et al, 2019, ice core CO2 data become very noisy in this interval. Although the exact mechanism remains open for debate, this is an important caveat when discussing the agreement or lack thereof with WAIS Divide.

I agree that both records capture the YD-PB jump in CO2 very nicely. But beyond this, it is difficult to say what variability is common to both records. For example, there is essentially one major oscillation in the WAIS Divide data to compare with in the Holocene (~10.6ka) that appears seems to be heavily biased by one very high CO2 value.

The WAIS Divide $CO_2$ data was reconstructed from the ice just below the bubble clathrate transition zone (BTCZ). Previous studies raised an issue about the possibility of high frequency noise of atmospheric $CO_2$ record in the ice just below the bubble clathrate transition zone (Lüthi et al., 2010; Shackleton et al, 2019). This phenomenon might be related to gas fractionation effect at the bubble-clathrate transformation depth zone. Gas content starts to be fractionated in the BCTZ because of the differential permeation of gas when bubbles have transformed to clathrates. $CO_2$ concentration in the first layer of clathrates is more enriched with higher bubble-to-clathrate permeation rates. Below the BCTZ, gas content slowly homogenizes again through molecular diffusion (Bereiter et al., 2009), which can cause high frequency noise to the ice below the BCTZ. Thus, part of the WAIS Divide $CO_2$ data may not be sufficient to discuss millennial variabilities of the early Holocene.

This is mentioned in the revised manuscript in 3.2 Comparison with existing $CO_2$ records for the early Holocene

On the other hand, comparison to the Dome C seems exceptionally difficult given the resolution of the data and the small amplitude. For example, Figure 2 implies that Dome C resolves variations, on the order of ~1 ppm, but the precision of the measurement at the time the data was produced is quoted as 1ppm (Monnin et al., 2001). As a test, it would be useful to show the correlation between the Siple Dome and all the other CO2 records to quantify the levels of agreement (alternatively, the differences could be plotted). At the moment, the discussion itself tends to be well balanced, but the conclusions reached seem to me to be too positive about the apparent agreement between the datasets. For example, it is stated quite a few times (using slightly different phrasing) "In conclusion, the existing ice core records support the millennial CO2 changes in the Siple Dome record although their temporal resolutions are not sufficient." This statement contains a contradiction. If the records are of insufficient resolution how can they be used to test the reliability of Siple Dome? My take-home from that section was the comparisons are inconclusive.

To verify the levels of agreement, we calculate the Pearson correlation coefficient between Siple Dome $CO_2$ and existing $CO_2$ records (Figure 3). Before calculating the correlation coefficient between Siple Dome $CO_2$ record and other $CO_2$ records, all $CO_2$ records were smoothed by 250-running means to eliminate high frequency noise. The offsets between existing $CO_2$ records and our data are also calculated (Figure 3).

We observe that $CO_2$ data sets from Siple dome and Dome C share similar trends in $CO_2$ variations despite the $CO_2$ offset in longer term means of 3~8 ppm. The $CO_2$ record from the Siple Dome is highly correlated with the Dome C record (r = 0.89, p < 0.001). The $CO_2$ offset between Dome C record and Siple Dome record decreases continuously from 11.7 ka to 7 ka with small variations at around 9.3 and 8.3 ka (Figure 3B). These small variations of $CO_2$ offset might be caused by the low sampling resolution and a stronger damping effect on the $CO_2$

concentration change due to the slower gas trapping process at Dome C (Spahni et al., 2003). The correlation coefficient between Siple Dome $CO_2$ and WAIS divide $CO_2$ is 0.7 ($p < 0.001$). However, the $CO_2$ offset between WAIS Divide and Siple Dome records is quite random (Figure 3B) because of scattering the WAIS Divide $CO_2$ record during the early Holocene period. Although the existing low resolution $CO_2$ records are not sufficient for comparison with our Siple Dome record, we cannot exclude the possibility of the millennial variations in the Siple Dome record due to the significant positive correlations.

[Figure]

**Figure 3.** A: Atmospheric $CO_2$ records. Red dots: Atmospheric $CO_2$ record from Dome C ice core. Red line: 250-running mean of atmospheric $CO_2$ record from Dome C ice core. Blue dots: Atmospheric $CO_2$ record from Siple Dome ice core. Blue line: 250-running mean of atmospheric $CO_2$ record from Siple Dome ice core. Green dots: Atmospheric $CO_2$ record from WAIS Divide ice core. Green line: 250-running mean of atmospheric $CO_2$ record from WAIS Divide ice core. B: $CO_2$ offset between Siple Dome $CO_2$ record and other published $CO_2$ records. Red line: $CO_2$ offset between Siple Dome $CO_2$ record and Dome C $CO_2$ record. Green line: $CO_2$ offset between Siple Dome $CO_2$ record and WAIS divide $CO_2$ record.

This is mentioned in the revised manuscript in 3.2 Comparison with existing $CO_2$ records for the early Holocene

The discussion of the CO2 offsets relies heavily on the findings in Nehrbass-Ahles's PhD thesis.  As far as I know this thesis is not publically available. For me, some parts of the discussion seem to assume that the reader could easily verify various conclusions in the thesis.

We mentioned Nehrbass-Ahles's PhD thesis to mention the largest $CO_2$ offset between Dome C and Law Dome records during the early Holocene. As the reviewer mentioned, the Law Dome record during the early Holocene is not published yet. Because it is not easy to discuss the $CO_2$ offset between Dome C and Law Dome records without showing the unpublished Law dome $CO_2$ record, we will delete the relevant words.

**Mechanistic links between solar forcing and CO2**

At the transition between the results and discussion there is missing logical step that links solar variability to the climate-driven mechanisms hypothesized to be responsible for carbon cycle changes. The authors have just highlighted in the results section that the most interesting finding is the correlation with solar proxies but then immediately jump into a discussion of various climate proxies without mentioning how solar variability could plausibly force these changes. Some of these links can be gleaned from information in the introduction but I would think it would better placed at the onset of the discussion.

I am sceptical that solar forcing could explain all this variability or, moreover, disentangled from all the forms of internal climate variability. I would welcome a clearer presentation of the chain of events/mechanisms that could link solar to CO2 (e.g. a schematic).

The driver of climate variations during the early Holocene is still unknown. However, many studies suggested periodicity of proxies and solar activities during the Early Holocene (11.5–7 ka) is similar (Liu et al., 2020). This implies that solar forcing (external forcing) might cause climate variabilities on millennial time scales during this period. We also observed maximum correlation coefficients between $CO_2$ and proxies with $CO_2$ time lags (Figure 4 and Table 2 in the manuscript). We assumed that high (lower) solar activity cause increased (decreased) atmospheric $CO_2$.

We found maximum correlation coefficients between $CO_2$ and proxies with $CO_2$ time lags. Thus, we assumed that atmospheric $CO_2$ might be affected by solar activities via climate (Figure 4 and Table 2). How solar variability causes these changes in atmospheric $CO_2$ concentrations/climate on millennial time scales are still the subject of debate. Changes of solar activities may impact on stratospheric ozone concentrations, which can change stratospheric and tropospheric circulation patterns (Meehl et al., 2009). Higher solar activity enhanced precipitation in the Intertropical Convergence Zone (ITCZ) and South Pacific Convergence Zone (SPCZ) (Meehl et al.,2009; van Loon et al., 2007). The intensified moisture at those areas increased trade wind strength and upwelling in the East Equatorial Pacific region. These conditions lead to Na Niña like climate states on millennial time scale (Marchitto et al.,2010). This ocean condition change in the East Equatorial Pacific has affected the North Atlantic (Darby et al., 2012).

Adde to section 4.1:

Understanding a link between climate variations and solar activity on millennial time scales during the early Holocene is important to decipher carbon cycle mechanisms. However, the climate mechanisms have not yet been deciphered. A possible mechanism is that changes of solar activities may impact on stratospheric ozone concentrations, which can change stratospheric and tropospheric circulation patterns (Meehl et al., 2009). Higher solar activity may enhance the precipitation in the Intertropical Convergence Zone (ITCZ) and South Pacific Convergence Zone (SPCZ) (Meehl et al.,2009; van Loon et al., 2007). Consequently, the intensified moisture at those areas would increase trade wind strength and upwelling in the East Equatorial Pacific region. These conditions would lead to Na Niña like climate states on millennial time scale (Marchitto et al., 2010). This change in the East Equatorial Pacific might have affected the North Atlantic (Darby et al., 2012).

In a future revision, I recommend using the TSI reconstructions. See Roth and Joos, 2013 and references therein for an overview of what I believe is state-of-the-art. Although the reconstructions come with their own host caveats and uncertainties it would be useful to consider if the variability in TSI is plausibly large enough to impact the climate and carbon cycle as the authors suggest.

Thank you for your suggestions. However, TSI record reconstructed by Roth and Joos (2013) covers only 10 ka. In this study, we used $^{14}$C records from tree rings and $^{10}$Be records from the ice core. $^{14}$C and $^{10}$Be concentrations are affected by Earth's geomagnetic field change on long-term scales. Marchitto et al. (2010) discussed that Earth's geomagnetic field change can be filtered by a high-pass filtering. Thus, it is likely that the high pass filtered $^{14}$C and $^{10}$Be records shown in the manuscript represent variations of solar irradiance.

**Line by line comments:**

Line 93: "We use a similar method to calculate the significance of this correlation against a random red-noise process. At each of the 1,000 steps, we use an AR(1) model (lag-1 auto regression) to fit the series."

What does "the series" refer to here?

Revised: data series

"Then, we calculate the percentage of correlations between the randomized synthetic series that are lower than the correlation coefficients of the real series to assess the significance of the correlation."

Can you relate this to the more traditionally reported p-value? Alternatively, at what values is a test significant?

Added

**Table 2.** Correlation between Siple Dome $CO_2$ record and climate proxy records. Column A shows correlation coefficients between $CO_2$ and proxies with $CO_2$ time lags. Column B shows correlation coefficients between $CO_2$ and proxies without $CO_2$ time lag. "With MC" are mean values from the simulations taking age uncertainties into account. "Without MC" is the classic calculation of correlation, without taking age uncertainty into account. Significance of the lag correlations was assessed against 1,000 repetitions of the lag correlation calculation using synthetic data stochastically generated to have the same red noise characteristics as the original series.

| Proxy records (Reference) | A: Correlation between $CO_2$ and proxies with $CO_2$ time lag (yrs) | | | | B: Correlation between $CO_2$ and proxies without $CO_2$ time lag | |
|---|---|---|---|---|---|---|
| | With MC | | Without MC | | With MC | Without MC |
| | r (p-value) | Time lag | r (p-value) | Time lag | r (p-value) | r (p-value) |
| $CO_2$ - [14]C production rate Marchitto et al.(2010); Reimer et al.(2004) | -0.49± 0.12 (0.3192) | -20±148 | - 0.76 (0.0003) | 50 | -0.48 (0.007) | -0.70 (< 0.001) |
| $CO_2$ - [10]Be flux from Greenland ice core  Finkel and Nishiizumi (1997); Marchitto et al. (2010); Vonmoos et al. (2006) | -0.52± 0.08 (0.2847) | 110±63 | - 0.61 (0.0087) | 110 | -0.29 (0.05) | -0.32 (< 0.001) |
| $CO_2$ - IRD from the North Atlantic region  Bond et al. (2001); Marchitto et al. (2010) | -0.49± 0.1 (0.3084) | 120±155 | - 0.73 (0.0009) | 170 | -0.33 (0.05) | -0.21 (< 0.001) |
| $CO_2$ - SST from eastern equatorial Pacific Marchitto et al. (2010) | -0.40± 0.13 (0.337) | 50±219 | - 0.61 (0.009) | 80 | -0.38 (0.04) | -0.55 (< 0.001) |
| $CO_2$ - Sea ice in the Southern Ocean  Nielsen et al. (2004) | -0.35± 0.17 (0.2899) | 190±228 | - 0.57 (0.0151) | 100 | -0.24 (0.17) | -0.48 (< 0.001) |
| $CO_2$ - SST in the Southern Ocean Nielsen et al. (2004) | 0.35± 0.17 (0.3070) | 52±228 | 0.57 (0.0144) | 30 | 0.35 (0.06) | 0.56 (< 0.001) |
| $CO_2$ - NGRIP $\delta^{18}O$ Rasmussen et al. (2006) | 0.21± 0.07 (0.2684) | -130±63 | 0.11 (0.3411) | 270 | 0.09 (0.5) | 0.06 (0.2) |

Section "3.1 The new high-resolution CO2 record during the early Holocene". Would this section read better if it followed immediately on the from the analytical methods section? The flow of the paper is a bit interrupted by the inclusions of the time series methods before the actual data description.

The section 2.3 is moved to supplement information.

Line 134 "In-situ production of CO2 cannot be ruled out but the effect should not greatly impact the offset between records from the different ice cores." This conclusion seems a bit premature, as you have not discussed organic production. Also, the remaining section seems to suggest that in-situ production is indeed the likely culprit for the offsets among cores.

The sentence is deleted.

Lines 143-153. This section was a little hard to follow as it mostly refers to a figure or set of figures in Nehrbass-Ahles that the reader cannot see. For example, the paper refers to a very large offset between EDC and Law Dome data in this interval which, to my knowledge, has never been published. While it is probably a good idea to reference this thesis as it seems has influenced the discussion, I would suggest keeping the references to specific data and conclusions therein to a minimum.

Thank you for this, as you mentioned Law dome $CO_2$ record during the early Holocene is not published yet. As, it is difficult to show the unpublished Law Dome $CO_2$ record, it would be better to delete this paragraph.

Line 163 "The nssCa can be produced in ice by the carbonate-acid reaction or transported as a dissolved form." This discussion could use some nuance about what we've actually measured in the ice when we report nssC. Elevated nssCa is not a sure sign you've had production by reaction with carbonate minerals, most the time it's increased dust delivery. What you'd really want to see is some sort of anomalous change in the ratio of soluble Ca (increase) to preserved carbonate minerals (decrease) - which at the moment we can't easily measure. One reasons being is that when we melt the cores, some of these Ca-rich minerals dissolve. The best resources to think through these issues are the original CFA papers by Anklin et al. 1995.

Revised to: To further evaluate the in-situ $CO_2$ production, we considered potential reactions. First, we compared the $CO_2$ with non-sea-salt Ca (nssCa) content in the ice to check the carbonate-acid reaction in the ice. The concentration of nssCa is mainly controlled by dust delivery but it also can be produced partially by the carbonate-acid reaction in ice. Thus, we examined the concentration of nssCa ion in the Siple Dome and Dome C ice. The nssCa records do not correlate well with the filtered millennial $CO_2$ variations in both Siple Dome (r = −0.33) and Dome C (r = 0.15) records during the early Holocene (Figures S2 and Figure S3 in SI). In addition, the nssCa trends in Dome C and Siple Dome ice do not agree (Figures S2 and Figure S3 in SI), but millennial $CO_2$ variations do. Second, we checked the $CO_2$ production by oxidation of organic compounds (e.g., $2H_2O_2 + HCHO \rightarrow 3H_2O + CO_2$) in ice (Tschumi and Stauffer, 2000). The Dome C site is located further from the ocean than Siple Dome and we therefore expect lower organic content in the Dome C ice. Concentrations of organic compounds at our sampling depths are not available. However, the concentration of oxidant $H_2O_2$ on the top 2.5−100 m in the Siple Dome core is below the detection limit of ~0.02 μM (McConnell, 1997), although 0.02 μM $H_2O_2$ still has potential to produce $CO_2$ and can increase the mixing ratio in bubbles by 5 ppm given sufficient supply of organic compounds (Ahn et al., 2004).

Line 165 "we pay attention to the observation" please consider a different phrasing.

Deleted

Line "In summary, CO2 data sets from different ice cores share similar trends in CO2 change despite offsets in longer term means of a few ppm. These offsets between the Siple dome CO2 record and others do not impact our conclusions."

If "trends" refers to the gradual decline in CO2, I feel you have made a convincing case. However, if "trends" refers to millennial-scale changes that this needs more support.

[Figure]

**Figure 4.** Red circles are Siple Dome ice core records during the early Holocene (11.7-7.4 ka). The black line indicates the average of 1,0000 times modified akima simulations showing an error-weighted average of the $CO_2$ record. The dark shaded indicies 2σ uncertainties calculated from modified akima simulations.

To assess variability at the millennial time scale, we evaluated 250-year running means and their uncertainties by using a Monte Carlo approach. Random sampling was made from a probability distribution for each measured value and its standard deviation. If a standard deviation was smaller than the mean external reproducibility of the measurement (0.87 ppm of 1σ), we used 0.87 ppm as the uncertainty of a measured value. Then, interpolation and resampling (1 yr) were applied to generate an evenly-spaced time series and to calculate the 250-year running means. We repeated this series of simulations 10,000 times and evaluated the mean of 250-year running means and its uncertainty (shown as 2σ in Figure 1). We used a modified Akima method using the built-in makima function in Matlab for the interpolation and changing the interpolation (or smoothing) method resulted in insignificant differences in the 250-year running means.

This is written to supplementary information.

Line 181 "The rapid CO2 increase at 11.7-11.3 ka might be associated with abrupt warming at the end of the last glacial termination (Marcott et al., 2014; Monnin et al., 2001)". I would argue that this is clear now but please note that the "abrupt warming" is restricted to Greenland and parts of the NH.

Revised:

The rapid $CO_2$ increase at 11.7–11.3 ka might be associated with abrupt warming in the North Atlantic and abrupt strengthening of Atlantic Meridional Overturning Circulation at the end of the last glacial termination (Marcott et al., 2014; Monnin et al., 2001).

Lines ~200.   All r-values need supporting significance values (preferably p-value style)

Revised. Please see Table 2 in this document.

Line 215.   The comparison with the ATS seems like tangent as once I looked into the SI it seems the comparison focus on the CO2 jump at the onset of the Holocene.   If ATS is central to the discussion I would suggest showing it in Figure 3.

This paragraph is deleted.

Line 220.   On the topic of the solubility pump, note that ~5 ppm changes in CO2 would require ~0.5 deg C changes in mean ocean temperature.   Is this plausible given the regional changes in SST you show?

A local SST record in the South East Atlantic polar front (Nielsen et al., 2004) shows ~1°C changes during the early Holocene on millennial time scales. However, it is hard to quantify the solubility pump with a local record. Mean ocean temperature record is needed to estimate the solubility pump effect in detail.

Figure 4.   It would be helpful to have an arrow indicating which direction shows the proxy leading CO2 and which directions shows the proxy lagging CO2.

Revised. We inserted "$CO_2$ lags" and "$CO_2$ leads" in the graphs so that the readers may recognize the sign of the $CO_2$lags.

[Figure]

**Figure 4 in the manuscript.** Correlation coefficients between $CO_2$ and proxies with $CO_2$ time lag calculated from Monte Carlo simulation. Vertical lines in black indicate zero time lag. Vertical lines in blue indicate maximum correlation coefficients between $CO_2$ and proxies with $CO_2$ time lag. A) $^{14}C$ production rate and atmospheric $CO_2$. B) $^{10}Be$ flux and atmospheric $CO_2$. C) SST in the eastern equatorial Pacific and atmospheric $CO_2$. D) IRD from the North Atlantic and atmospheric CO2. E) SST in the East Equatorial Pacific indicating El Niño–like or La Niña–like conditions and atmospheric CO2. F) SI in the East Equatorial Pacific and atmospheric $CO_2$.

Sincerely,

Thomas Bauska

**References:**

Martin Anklin, Jean-Marc Barnola, Jakob Schwander, Bernhard Stauffer & Dominique Raynaud(1995) Processes affecting the CO2 concentrations measured in Greenland ice, Tellus B: Chemical and Physical Meteorology, 47:4, 461-470

Elsig, J., Schmitt, J., Leuenberger, D. et al. Stable isotope constraints on Holocene carbon cycle changes from an Antarctic ice core. Nature 461, 507–510 (2009). https://doi.org/10.1038/nature08393

Kaufman, D., McKay, N., Routson, C. et al. Holocene global mean surface temperature, a multi-method reconstruction approach. Sci Data 7, 201 (2020). https://doi.org/10.1038/s41597-020-0530-7

Lüthi, D., Bereiter, B., Stauffer, B., Winkler, R., Schwander, J., Kindler, P., Leuenberger, M., Kipfstuhl, S., Capron, E., Landais, A., Fischer, H., & Stocker, T. F. (2010). CO2 and O2/N2 variations in and just below the bubble-clathrate transformation zone of Antarctic ice cores. Earth and Planetary Science Letters, 297(1–2), 226–233.

Marsicek, J., Shuman, B. N., Bartlein, P. J., Shafer, S. L. & Brewer, S. Reconciling divergent trends and millennial variations in Holocene temperatures. Nature 554, 92 (2018).

Roth, R. and Joos, F.: A reconstruction of radiocarbon production and total solar irradiance from the Holocene 14C and CO2 records: implications of data and model uncertainties, Clim. Past, 9, 1879–1909, https://doi.org/10.5194/cp-9-1879-2013, 2013.

Shackleton, S., Bereiter, B., Baggenstos, D., Bauska, T. K., Brook, E. J., Marcott, S. A., & Severinghaus, J. P. (2019) Is the noble gas based rate of ocean warming during the Younger Dryas overestimated? Geophysical Research Letters, 46, 5928-5936. https://doi.org/10.1029/2019GL082971

**Citation**: https://doi.org/10.5194/cp-2021-113-RC2

**References**

Ahn, J., Brook, E. J., Mitchell, L., Rosen, J., McConnell, J. R., Taylor, K., Etheridge, D., and Rubino, M.: Atmospheric $CO_2$ over the last 1000 years: A high-resolution record from the West Antarctic Ice Sheet (WAIS) Divide ice core, Global Biogeochem. Cy., 26, GB2027, https://doi.org/10.1029/2011GB004247, 2012.

Anklin, M., Barnola, J.-M., Schwander, J., Stauffer, B., and Raynaud, D.: Processes affecting the $CO_2$ concentration measured in Greenland ice, Tellus Ser., B(47), 461–470, 1995.

Lean, J.: Evolution of the sun's spectral irradiance since the Maunder Minimum, Geophys. Res. Lett., 27, 2425–2428, 2000.

Liu, X., Sun, Y., Vandenberghe, J., Cheng, P., Zhang, X., Gowan, E. J., Lohmann, G., and An, Z.: Centennial- to millennial-scale monsoon changes since the last deglaciation linked to solar activities and North Atlantic cooling, Clim. Past, 16, 315–324, https://doi.org/10.5194/cp-16-315-2020, 2020.

Meehl, G. A., Arblaster, J. M., Matthes, K., Sassi, F., and van Loon, H.: Amplifying the Pacific Climate System Response to a Small 11-Year Solar Cycle Forcing, Science, 325, 1114–1118, 2009

Monnin, E., Steig, E. J., Siegenthaler, U., Kawamura, K., Schwander, J., Stauffer, B., Stocker, T. F., Morse, D. C., Barnola, J.- M., Bellier, B., Raynaud, D., and Fischer, H.: Evidence for substantial accumulation rate variability in Antarctica during the Holocene through synchronization of $CO_2$ in the Taylor Dome, Dome C and DML ice cores, Earth Planet. Sc. Lett., 224, 45–54, 2004a.

Monnin, E., Steig, E. J., Siegenthaler, U., Kawamura, K., Schwander, J., Stauffer, B., Stocker, T. F., Morse, D. C., Barnola, J.- M., Bellier, B., Raynaud, D., and Fischer, H.: EPICA Dome C ice core high resolution Holocene and transition $CO_2$ data, Technical report, IGBP PAGES/World Data Center for Paleoclimatology, OAA/NGDC Paleoclimatology Program, Boulder CO, USA, 2004b.

Monnin, E., Indermühle, A., Dällenbach, A., Flückiger, J., Stauffer, B., Stocker, T. F., Raynaud, D., and Barnola, J. M.: Atmospheric $CO_2$ concentrations over the last glacial termination, Science, 291, 112–114, doi:10.1126/science.291.5501.112, 2001.

Roth, R. and Joos, F.: A reconstruction of radiocarbon production and total solar irradiance from the Holocene $^{14}C$ and $CO_2$ records: implications of data and model uncertainties, Clim. Past, 9, 1879–1909, https://doi.org/10.5194/cp-9-1879-2013, 2013.

Rubino, M., Etheridge, D. M., Trudinger, C. M., Allison, C. E., Battle, M. O., Langenfelds, R. L., Steele, L. P., Curran, M., Bender, M., White, J. W. C., Jenk, T. M., Blunier, T., and Francey, R. J.: A revised 1000-year atmospheric $\delta^{13}C$- $CO_2$ record from Law Dome and South Pole, Antarctica, J. Geophys. Res.-Atmos., 118, 8482–8499, doi:10.1002/jgrd.50668, 2013.

Rubino, M., Etheridge, D. M., Thornton, D. P., Howden, R., Allison, C. E., Francey, R. J., Langenfelds, R. L., Steele, L. P., Trudinger, C. M., Spencer, D. A., Curran, M. A. J., van Ommen, T. D., and Smith, A. M.: Revised records of atmospheric trace gases $CO_2$, $CH_4$, $N_2O$, and $\delta^{13}C$-$CO_2$ over the last 2000 years from Law Dome, Antarctica, Earth Syst. Sci. Data, 11, 473–492, https://doi.org/10.5194/essd-11-473-2019, 2019.

Siegenthaler, U., Monnin, E., Kawamura, K., Spahni, R., Schwander, J., Stauffer, B., Stocker, T. F., Barnola, J.- M., and Fischer, H.: Supporting evidence from the EPICA Dronning Maud Land ice core for atmospheric $CO_2$ changes during the past millennium, Tellus B, 57(7), 51–57, doi:10.1111/j.1600- 0889.2005.00131.x, 2005.

Shackleton, S., Bereiter, B., Baggenstos, D., Bauska, T. K., Brook, E. J., Marcott, S. A., and Severinghaus, J. P.: Is the Noble Gas-Based Rate of Ocean Warming During the Younger Dryas Overestimated?, Geophys. Res. Lett., 46, 5928–5936, https://doi.org/10.1029/2019GL082971, 2019.

Varma, V., Prange, M., Lamy, F., Merkel, U., and Schulz, M.: Solar-forced shifts of the Southern Hemisphere Westerlies during the Holocene, Clim. Past, 7, 339–347, https://doi.org/10.5194/cp-7-339-2011, 2011.

---

## Editor Decision (ED1)

Thank you for your revised paper, which I sent to two experts for re-review. Firstly I add my apologies that the process took so long: I was keen to get the opinion of both original reviewers but this took a lot longer than anticipated.

The result of the re-review is rather unsatisfying. Both reviewers want to see your data published as an important addition to the overall CO2 dataset from ice cores. However both of them felt you had not really addressed their primary concerns (1) about how robust the variations you discuss are and (2) about whether you have made a convincing case for a solar forcing of the variations.

I have therefore decided that I will in principle consider a further revision with a view to publishing the data. I have classed it as minor revision because I do not intend to ask for further review: I know what the reviewers are worried about and I will be able to judge whether you have dealt with the issues to an acceptable level. I thus want to make it clear that while this is technically classed as minor revision, I am expecting substantial changes (as outlined below), and the paper could still be rejected after the next revision if I don't see them.

Please consider all the comments made by the reviewers. I will now explain what I see as the main issues that still need to be addressed. This includes some minor issues I have noted myself. Please answer each of my comments as well as those of the reviewers.

1. Data quality. In their para 2, rev 1 makes the point that many of the individual data points show a much larger error bar than the cited analytical uncertainty. Please address this point, as it is clearly not the case that the value at a particular depth is known to within 0.87 ppm.

2. I think the issue about the very narrow error envelope shown in Fig 1 and FigS6 is about presentation. This is the envelope of 250-year averages. But because it is shown continuously it gives the impression that even centennial scale wiggles in the data are real, which is not defendable. Please deal with this at minimum by the following:
(a) On Fig 1, the caption  please add "2 sigma uncertainties of the 250-year mean value, and cannot be used to interpret variations on shorter timescales" .
(b) I am very concerned that the minima and maxima you later interpret might be strongly influenced by single outliers, eg at 10.8 ka and 11.1 ka. By using 250 year means you are implicitly assuming that the true concentration is rather smooth, and that the existence of data points far from the smooth line is the result of deviations caused by the enclosure process and that these are Gaussian. The existence of these outliers questions that.  Please carry out (maybe in supplement) a kind of bootstrap analysis. What I mean is that you should remove outliers (e.g. any data point more than a standard 2 sigma from the line) and show what the smoothed line then looks like. If this removes any of the major deviations you subsequently interpret than this should be stated in the text and should make your interpretation more cautious.
(c) A small technical point. In the text it says you did 10000 MonteCarlo runs, in the caption to fig 1 it says 1000. Please correct.

3. A second issue with data quality concerns the comparison with EDC and WAIS. There are some technical issues with this, as well as some opportunities missed.
(a) the text in lines 114-117 says "the CO2 offset between Dome C record and Siple Dome record is quite random (Figure 2B) because of scattering in the WAIS Divide". I assume you mean the offset between WAIS Divide and Siple Dome, please correct.
(b) It makes no sense to calculate correlations that include the major rise out of the YD. Of course all records will show correlations if they include a giant step. Please redo your correlations using only the period to 11.5 ka (the period you use in your filtered record in Fig 1).

(c) before interpreting very small variations in CO2 it is important to show they are robust, ie observed at different sites. As rev 2 says this will really only be tested when we have other data, but you can do more with what you have. You already dismissthe WAIS Divide data but you don't actually let the reader see the crucial comparison even for EDC-SD. Thus the reader cannot judge whether your statement "We observe that CO2 data sets from Siple Dome and Dome C share similar trends in CO2 variations" is correct. So please add a figure (I would propose in the main text (not supplement), maybe as another panel to Fig 2) in which you produce the filtered record (as in Fig 1B) for all 3 sites. When you have done this please discuss seriously how robust your findings are, and exercise an appropriate caution in the rest of the paper depending on the result.

(d) Line 164 and line 15 "The Siple Dome CO2 record shows millennial variability of ~2–6 ppm". Looking at Fig 1B, the maximum variation is clearly only 4 ppm, please correct.

4. The comparison with other records is OK to make (Fig 3) as long as you have caveated about how robust the variations you see are (as per my previous comments). However again please be honest and cautious. While you get reasonable correlations with 14C and 10be, it is nonetheless the case that only 2 of your 3 serious dips have an expression in your solar proxies. At 9.1k, the solar proxies are antiphased with CO2. You should mention this. Taken together with the discussion of later solar variations (around line 250 and discussed by rev 2), these should cause you to caution that the link with solar is very speculative.

5. Please redraft section 4 and the abstract to be very cautious based on all the above. In particular the sentence "These relationships suggest that weak solar forcing changes might have impacted CO2 by changing CO2 outgassing from the Southern Ocean and the East Equatorial Pacific and terrestrial carbon storage in the Northern Hemisphere over the early Holocene" suggests you have established a mechanism which is not the case. I am OK with you making the case that there is a tentative correlation with solar forcing but in the abstract you should not go further.

---

## Author Response (AR2)

Dear editor and reviewers,

We thank two reviewers and the editor for their careful review of our paper, and their suggestions. Our detailed responses to the comments are shown in blue, and the resulting changes in the manuscript are shown in green.

On behalf of all co-authors,

Jinhwa Shin

**Anonymous Referee #1**

The authors present revisions to the original manuscript and address comments by two reviewers. I appreciate the detailed response to the comments. Again, the new high-resolution data of the Siple Dome record is a very valuable data set for the period 7.4 to 11.7 kyrBP. This data set should eventually be available to the scientific community. However, the revised version still suffers from the two major issues: (i) statistical treatment of the data to isolate millennial-scale variability, (ii) interpretation of the purported cyclic variations found based on (i).

Many minor points were addressed by the authors in the revised version, but the two major issues are still of concern and are not convincingly addressed. This prevents me from recommending publication.

Major Comments:

1) The uncertainty band of the CO2 data remains unrealistically narrow as Figure 1 evidences. a 2*sigma uncertainty of only 0.87 ppm may result from a statistical analysis but does not withstand common-sense analysis of the original data as depicted in Fig. 1.

The uncertainty value, 0.87 ppm, is obtained by Monte Carlo simulation. Due to a 250-running mean, those small variations were removed, thus, $2\sigma$ uncertainties calculated from Monte Carlo simulations is 0.87 ppm.
According to the individual $CO_2$ data points, the standard error of the mean of replicates from the same depth interval was 0.8 and 0.5 ppm on average for SNU and OSU data, respectively. The range of the standard error is from 0.01 to 1.75 ppm (1σ). To make it clear, a sentence and the caption are added in section 3.1 and on Figure 1 respectively.

A sentence added to section 3.1: ranging from 0.01 to 1.75 ppm.
The caption added on Figure 1: "2 sigma uncertainties of the 250-year mean value, and cannot be used to interpret variations on shorter timescales".

A 2*sigma uncertainty band should contain about 95% of the values, when assuming Gaussian distribution of the uncertainty. The number of data points outside the grey band in Fig. 1 is clearly exceeding 5% of the total number of data points.

[Figure]

**Figure R1.** Red dots indicate atmospheric $CO_2$ records obtained from Siple Dome ice core. Grey band indicates 1σ envelop by using Monte Carlo average

We calculate 1σ envelop by using Monte Carlo average. There are two outliers at 11.08 and 10.83 ka. Thus, we conducted a high-pass filtering at 1/1800 year$^{-1}$ without two single outliers at 11.08 and 10.83 ka. The trend of $CO_2$ data filtered by high pass filtering without 2 points at 11.8 ka and 10.83 ka is similar to the trend of original data filtered by high pass filtering at 1/1800 year$^{-1}$ (Figure R2). In addition, the magnitude of $CO_2$ variation at around 11 ka becomes smaller but these two records show the almost same local minima.

[Figure]

**Figure R2.** A. Green line indicates $CO_2$ original data which was filtered by high pass filtering. B. Blue line indicates $CO_2$ data filtered by high pass filtering without 2 points at 11.8 ka and 10.825 ka.

We also calculate the correlation coefficient between $CO_2$ without two outliers at 10.8 ka and 11.1 ka and climate proxies (Table R1). These results are almost similar to the results calculated with the original $CO_2$ record. In my

opinion, these two outliers at 11.08 and 10.83 ka may not highly impact our interpretation. Figure R2 and Table R1 were added in the Supplement.

**Table R1.** Correlation between Siple Dome $CO_2$ record without single outliers at 10.8 ka and 11.1 ka and climate proxy records. Column A shows correlation coefficients between $CO_2$ and proxies with $CO_2$ time lags. Column B shows correlation coefficients between $CO_2$ and proxies without $CO_2$ time lag. "With MC" are mean values from the simulations taking age uncertainties into account. "Without MC" is the classic calculation of correlation, without taking age uncertainty into account. Significance of the lag correlations was assessed against 1,000 repetitions of the lag correlation calculation using synthetic data stochastically generated to have the same red noise characteristics as the original series.

| Proxy records | A: Correlation between $CO_2$ and proxies with $CO_2$ time lag (yrs) | | | | B: Correlation between $CO_2$ and proxies without $CO_2$ time lag | |
| --- | --- | --- | --- | --- | --- | --- |
| | With MC | | Without MC | | With MC | Without MC |
| | r (p-value) | Time lag | r (p-value) | Time lag | r (p-value) | r (p-value) |
| $CO_2$ - $^{14}C$ production rate Marchitto et al.(2010); Reimer et al.(2004) | -0.44±0.10 (0.010) | 0±148 | -0.76 (<0.001) | 40 | -0.43 (0.005) | -0.62 (<0.001) |
| $CO_2$ - $^{10}Be$ flux from Greenland ice core Finkel and Nishiizumi (1997); Marchitto et al. (2010); Vonmoos et al. (2006) | -0.30±0.06 (0.101) | 130±63 | -0.58 (<0.001) | 120 | -0.30 (0.021) | -0.36 (<0.001) |
| $CO_2$ - IRD from the North Atlantic region Bond et al. (2001); Marchitto et al. (2010) | -0.44±0.11 (0.076) | 70±155 | -0.73 (<0.001) | 160 | -0.32 (0.057) | -0.23 (0.001) |
| $CO_2$ - SST from eastern equatorial Pacific Marchitto et al. (2010) | -0.37±0.13 (0.057) | 0±219 | -0.61 (<0.001) | 80 | -0.34 (0.044) | -0.56 (<0.001) |
| $CO_2$ - Sea ice in the Southern Ocean Nielsen et al. (2004) | -0.32±0.16 (0.171) | -180±228 | -0.57 (<0.001) | 80 | -0.24 (0.155) | -0.49 (<0.001) |
| $CO_2$ - SST in the Southern Ocean Nielsen et al. (2004) | 0.35±0.16 (0.075) | 60±228 | 0.58 (<0.001) | 20 | 0.35 (0.063) | 0.58 (<0.001) |
| $CO_2$ - NGRIP $\delta^{18}O$ Rasmussen et al. (2006) | 0.18±0.06 (0.180) | -140±63 | 0.20 (0.080) | -110 | 0.17 (0.180) | 0.16 (0.001) |

A paragraph added to section 3.3: There are two outliers at ~11.08 and 10.83 ka, which are far from the 250-running mean. The Siple Dome $CO_2$ record except for the two data points at ~11.08 and 10.83 ka was smoothed and high pass filtered at 1/1800 yr. With this processed data, we calculated correlation coefficients between the filtered $CO_2$ and climate proxy series again (Figure S6 and Table S3). The correlation coefficients between climate proxies and $CO_2$ data except for two outliers at ~11.08 and 10.83 ka are similar to those between the original $CO_2$ record and the climate proxies, showing that the two outliers may not highly impact our interpretation.

2)  The authors quote 0.87 ppm as "the uncertainty of a measured value". This cannot be true as many of the data points in Fig. 1 indicate an individual uncertainty of 3 ppm, a much more realistic value of the current analytical approaches to measure CO2 on ice samples. This point is a fundamental one for it determines to which extent variations in the time series can be identified.

The uncertainty value, 0.87 ppm is obtained by Monte Carlo simulation, which means that this value is not individual uncertainty. The main reason of the small uncertainty is attributed to the removal of the high frequency signal by a 250-running mean.

3)  The successive filtering – 250-year running mean, then 1/(1800 year high-pass filtering – with a 1-year interpolation in between these steps is still applied. The authors essentially employ a band-pass filter to isolate variability on time scales from 250 to 1800 years. This treatment will necessarily result in some millennial variability even if applied to a white noise time series. I am therefore not convinced that this statistical analysis provides an unbiased view of the CO2 data. Therefore, I remain very skeptical of the treatment of the data.

This statistic is wildly used in paleoclimatology (Marchitto et al., 2010; Schmidt et al., 2012; Yang et al., 2017. ). As you concerned, this statistic tool can create artificial variations or delete small variations. However, as mentioned in the previous response letter, all the original records as well as 250-year running means were presented. The original proxy and $CO_2$ records clearly show the millennial variations. Thus, the treatment may not affect our conclusions. Additional $CO_2$ concentration using better-quality ice cores and carbon cycle models will be very helpful to confirm the observation.

4) The absence of a mechanistic model, or at least causal chain, to relate solar variability to CO2 changes makes the statements very speculative. Unless a stronger case is built I am seriously doubting the conclusions of this paper as formulated in the last sentence of the abstract.

**Abstract revised to:**

We present a new high-resolution record of atmospheric $CO_2$ from the Siple Dome ice core, Antarctica over the early Holocene (11.7–7.4 ka) that quantifies natural $CO_2$ variability on millennial timescales under interglacial climate conditions. Atmospheric $CO_2$ decreased by ~10 ppm between 11.3 and 7.3 ka. The decrease was punctuated by local minima at 11.1, 10.1, 9.1 and 8.3 ka with amplitude of 2–4 ppm. Although the linkage between atmospheric $CO_2$ and the climate change remains uncertain due to insufficient paleoclimate records and model simulations, these variations correlate with proxies for solar forcing and local climate in the South East Atlantic polar front, East Equatorial Pacific and North Atlantic. Additional $CO_2$ measurements from a higher accumulation site and carbon cycle models are needed.

<end of review>

First and foremost, I profusely apologize for being so late with this review. In hindsight, a part time work schedule meant that I should have managed my time differently with this paper.

Overall, the authors have addressed many the technical concerns of both reviewers regarding the data quality and processing. They have improved the description of the signal smoothing techniques and the question of offsets between $CO_2$ records is now treated more thoroughly. To the question of how accurate the reconstruction is presented by the authors, I am now of the opinion that the only true test of the data quality and data processing techniques will be done through replication studies using more precise methods and perhaps better-quality ice cores. It remains my opinion that the data set itself will be of wide interest to the paleoclimate community and is thus highly deserving of publication.

Regarding the mechanisms proposed in the study, both reviewers questioned the links to solar variability in the initial round. Ultimately, little has changed in the terms of the conclusions of the paper (note the abstract and conclusions remained relatively untouched). This is of course the prerogative of the authors, but I would have appreciated seeing a more in-depth rebuttal in the authors' responses. I personally would have thought the link to TSI would have featured less prominently in lieu of an expanded discussion on the direct carbon cycle mechanisms (terrestrial biosphere, Southern Ocean controls).

I suggested a test using the TSI and the late Holocene data that might allow the authors to test their solar link, which was partially completed. I was hoping that a nuanced discussion of how TSI can be highly variably in the last millennium, but $CO_2$ varies more slowly (possibly related to anthropogenic land-use changes) might temper the conclusions about solar forcing of $CO_2$. In the current manuscript, it seems that the test agreed with their hypothesis in one instance (Maunder) and not in the other considering all the cores and much more muted response in WAIS Divide (Sporer). Other oscillations in TSI were ignored (minima at ~1300 CE and ~1050CE) and there was no discussion of the lead/lag timing. For example, it seems strange of me that the sharpest drop in $CO_2$ of the last millennium (~1600 CE) coincides with a maximum in TSI and not a minimum. With only two tests and a 50% success rate, the results are most likely down to random chance. In my opinion, the paper doesn't really follow through on these results into the discussion and conclusions and they are simply presented as a result. Again, in my opinion, there is a conclusion to be arrived from this, which is that whilst solar forcing might be important in some instances, it is clearly not the major driver of all sub-millennial $CO_2$ variability. If the author's feel the test is inconclusive than it might be a better choice not to include it at all rather than include it but not tie it to any firm conclusions.

We fully agree with your suggestions. Discussing the relationship between atmospheric $CO_2$ and solar forcing on shorter time scales during the late Holocene would be great to understand the major driver of $CO_2$ variabilities during the early Holocene. However, it is not easy to compare atmospheric $CO_2$ during the early Holocene with atmospheric $CO_2$ during the Late Holocene due to different boundary conditions during the early Holocene and the late Holocene. For example, variations of solar forcing are large on a centennial time scale during the Early Holocene. Thus, the solar output effect might be enhanced since the climate system is not responded linearly (Mohtadi et al., 2016). However, due to a decrease in summer insolation and the small variation of solar forcing during the late Holocene (7–1 ka) (Berger, 1978), solar forcing might play a less important role during the late Holocene.

Atmospheric $CO_2$ can be controlled by $CO_2$ exchange between the ocean and the atmosphere, as well as changes of terrestrial carbon stocks. Each reservoir has different response time scales. The deep ocean inventory requires a few millennia to re-equilibrate to climate change (Schmittner and Galbraith, 2008). However, the response of the terrestrial biosphere is usually fast (decadal to centennial timescale) (Bouttes et al., 2012; Menviel et al., 2014; Schmittner and Galbraith, 2008). Thus, atmospheric $CO_2$ might be affected by changes in solar forcing via the terrestrial processes on the short time scales. However, it is difficult to investigate the relationship between $CO_2$ and solar forcing (or TSI) during 1900–1000 CE due to anthropogenic causes such as wars and pandemic diseases. Additional studies are needed.

**Section 4 revised to**: In this study, we observed that atmospheric $CO_2$ is highly anti-correlated with the [14]C production rate and [10]Be flux with $CO_2$ time lag during the early Holocene (Figure 3). However it is the case that large variations of solar forcing at ~11.1, 10.1 and 8.3 ka. The [14]C production rate and [10]Be flux are correlated with $CO_2$ at ~9.1 ka on submillennial time scales.

We also check the correlation of $CO_2$ with solar activity during the last 2000 years on centennial time scales. A positive correlation between solar forcing and atmospheric $CO_2$ is observed during the Little Ice Age (LIA). There are two periods in which sunspots were exceedingly rare. During the Maunder sunspot minimum (1647–1715 CE), total solar irradiance (TSI) was reduced by $0.85\pm0.16$ W m$^{-2}$. Atmospheric $CO_2$ records from Antarctic ice cores commonly show a decrease trend during this period (Ahn et al., 2012; Monnin et al., 2004; Siegenthaler et al., 2005; Rubino et al., 2019). During the Spörer Minimum (1450–1550 CE), TSI record during this period also shows a decrease trend. However, atmospheric $CO_2$ decrease is not significant in Law Dome and EPICA Dronning Maud Land (EDML) records (Monnin et al., 2004; Siegenthaler et al., 2005; Rubino et al., 2019), while WAIS divide ice record shows a decrease during this period (Ahn et al., 2012) (Figure S7 in SI). However, atmospheric $CO_2$ decrease drastically at ~1600 CE when total solar irradiance (TSI) shows a local maximum, which is similar to the relationship between solar forcing and atmospheric $CO_2$ at ~9.1 ka. To conclude, it is vague how solar forcing is related with atmospheric $CO_2$ variations on millennial time scales. Comparing the early and last Holocene requires attention due to different boundary conditions during these two periods and anthropogenic $CO_2$ during the late Holocene (e.g., Ruddiman, 2003, 2007). Variations of solar forcing are large on a centennial time scale during the Early Holocene. Thus, the solar output effect might be enhanced since the climate system is not responded linearly (Mohtadi et al., 2016). However, due to a decrease in summer insolation and the small variation of solar forcing during the late Holocene (7–1 ka) (Berger, 1978), solar forcing might play a less important role during the late Holocene. Further studies are needed to understand the relationship between atmospheric $CO_2$ and solar forcing on shorter time scales during the early Holocene with more proxy records and numerical models.
.

My recommendation is that paper can now be published in its current form with the understanding that further refinement of the underlying ice core records is needed with more data and that a discussion of the solar links will be borne out in the subsequent literature.

Sincerely,
Thomas Bauska

Thank you for your revised paper, which I sent to two experts for re-review. Firstly I add my apologies that the process took so long: I was keen to get the opinion of both original reviewers but this took a lot longer than anticipated.

The result of the re-review is rather unsatisfying. Both reviewers want to see your data published as an important addition to the overall CO2 dataset from ice cores. However both of them felt you had not really addressed their primary concerns (1) about how robust the variations you discuss are and (2) about whether you have made a convincing case for a solar forcing of the variations.

I have therefore decided that I will in principle consider a further revision with a view to publishing the data. I have classed it as minor revision because I do not intend to ask for further review: I know what the reviewers are worried about and I will be able to judge whether you have dealt with the issues to an acceptable level. I thus want to make it clear that while this is technically classed as minor revision, I am expecting substantial changes (as outlined below), and the paper could still be rejected after the next revision if I don't see them.

Please consider all the comments made by the reviewers. I will now explain what I see as the main issues that still need to be addressed. This includes some minor issues I have noted myself. Please answer each of my comments as well as those of the reviewers.

1. Data quality. In their para 2, rev 1 makes the point that many of the individual data points show a much larger error bar than the cited analytical uncertainty. Please address this point, as it is clearly not the case that the value at a particular depth is known to within 0.87 ppm.

Sentence added in Section 3.1: ranging from 0.01 to 1.75 ppm

2. I think the issue about the very narrow error envelope shown in Fig 1 and FigS6 is about presentation. This is the envelope of 250-year averages. But because it is shown continuously it gives the impression that even centennial scale wiggles in the data are real, which is not defendible. Please deal with this at minimum by the following:

(a) On Fig 1, the caption please add "2 sigma uncertainties of the 250-year mean value, and cannot be used to interpret variations on shorter timescales" .
Added

(b) I am very concerned that the minima and maxima you later interpret might be strongly influenced by single outliers, eg at 10.8 ka and 11.1 ka. By using 250 year means you are implicitly assuming that the true concentration is rather smooth, and that the existence of data points far from the smooth line is the result of deviations caused

by the enclosure process and that these are Gaussian. The existence of these outliers questions that. Please carry out (maybe in supplement) a kind of bootstrap analysis. What I mean is that you should remove outliers (e.g. any data point more than a standard 2 sigma from the line) and show what the smoothed line then looks like. If this removes any of the major deviations you subsequently interpret than this should be stated in the text and should make your interpretation more cautious.

We conducted a high-pass filtering at 1/1800 year$^{-1}$ without two single outliers at 11.08 and 10.825 ka. The trend of $CO_2$ data filtered by high pass filtering without 2 points at 11.8 ka and 10.825 ka is similar to the trend of original data filtered by high pass filtering at 1/1800 year$^{-1}$ (Figure 1 in this document). In addition, the magnitude of $CO_2$ variation at around 11 ka becomes smaller but these two records show the almost same local minima.

[Figure]

**Figure R2.** A. Green line indicates $CO_2$ original data which was filtered by high pass filtering. B. Blue line indicates $CO_2$ data filtered by high pass filtering without 2 points at 11.8 ka and 10.825 ka.

We also calculated the correlation coefficient between $CO_2$ without two outliers at 10.8 ka and 11.1 ka and climate proxies (Table R1). These results are almost similar to the results calculated with the original $CO_2$ record. In my opinion, these two outliers at 10.8 ka and 11.1 ka may not highly impact our interpretation. Figure R1 and Table R1 were added in the Supplement.

**Table R1.** Correlation between Siple Dome $CO_2$ record without outliers at 10.8 ka and 11.1 ka and climate proxy records. Column A shows correlation coefficients between $CO_2$ and proxies with $CO_2$ time lags. Column B shows correlation coefficients between $CO_2$ and proxies without $CO_2$ time lag. "With MC" are mean values from the simulations taking age uncertainties into account. "Without MC" is the classic calculation of correlation, without taking age uncertainty into account. Significance of the lag correlations was assessed against 1,000 repetitions of the lag correlation calculation using synthetic data stochastically generated to have the same red noise characteristics as the original series.

| Proxy records | A: Correlation between $CO_2$ and proxies with $CO_2$ time lag (yrs) | | | | B: Correlation between $CO_2$ and proxies without $CO_2$ time lag | |
|---|---|---|---|---|---|---|
| | With MC | | Without MC | | With MC | Without MC |
| | r (p-value) | Time lag | r (p-value) | Time lag | r (p-value) | r (p-value) |
| $CO_2$ - $^{14}$C production rate
Marchitto et al.(2010);
Reimer et al.(2004) | -0.44±0.10
(0.010) | 0±148 | -0.76
(<0.001) | 40 | -0.43
(0.005) | -0.62
(<0.001) |
| $CO_2$ - $^{10}$Be flux from Greenland ice core
Finkel and Nishiizumi (1997);
Marchitto et al. (2010);
Vonmoos et al. (2006) | -0.30±0.06
(0.101) | 130±63 | -0.58
(<0.001) | 120 | -0.30
(0.021) | -0.36
(<0.001) |
| $CO_2$ - IRD from the North Atlantic region
Bond et al. (2001);
Marchitto et al. (2010) | -0.44±0.11
(0.076) | 70±155 | -0.73
(<0.001) | 160 | -0.32
(0.057) | -0.23
(0.001) |
| $CO_2$ - SST from eastern equatorial Pacific
Marchitto et al. (2010) | -0.37±0.13
(0.057) | 0±219 | -0.61
(<0.001) | 80 | -0.34
(0.044) | -0.56
(<0.001) |
| $CO_2$ - Sea ice in the Southern Ocean
Nielsen et al. (2004) | -0.32±0.16
(0.171) | -180±228 | -0.57
(<0.001) | 80 | -0.24
(0.155) | -0.49
(<0.001) |
| $CO_2$ - SST in the Southern Ocean
Nielsen et al. (2004) | 0.35±0.16
(0.075) | 60±228 | 0.58
(<0.001) | 20 | 0.35
(0.063) | 0.58
(<0.001) |
| $CO_2$ - NGRIP $\delta^{18}$O
Rasmussen et al. (2006) | 0.18±0.06
(0.180) | -140±63 | 0.20
(0.080) | -110 | 0.17
(0.180) | 0.16
(0.001) |

A paragraph added to section 3.3: There are two outliers at ~11.08 and 10.83 ka, which are far from the 250-running mean. The Siple Dome $CO_2$ record except for the two data points at ~11.08 and 10.83 ka was smoothed and high pass filtered at 1/1800 yr. With this processed data, we calculated correlation coefficients between the

filtered $CO_2$ and climate proxy series again (Figure S6 and Table S3). The correlation coefficients between $CO_2$ data except for the two outliers at ~11.08 and 10.83 ka and climate proxies is similar to the relationship between original $CO_2$ record and climate proxies, which shows that two outliers may not highly impact our interpretation.

Figure S6 and Table S3 were added in Supplementary Information.

(c) A small technical point. In the text it says you did 10000 MonteCarlo runs, in the caption to fig 1 it says 1000. Please correct.
The caption was revised

3. A second issue with data quality concerns the comparison with EDC and WAIS. There are some technical issues with this, as well as some opportunities missed.

(a) the text in lines 114-117 says "the CO2 offset between Dome C record and Siple Dome record is quite random (Figure 2B) because of scattering in the WAIS Divide". I assume you mean the offset between WAIS Divide and Siple Dome, please correct.
Revised

(b) It makes no sense to calculate correlations that include the major rise out of the YD. Of course all records will show correlations if they include a giant step. Please redo your correlations using only the period to 11.5 ka (the period you use in your filtered record in Fig 1).
Figure 3 shows data from 12 ka to 7 ka. However, to calculate correlations, we only used filtered data from 11.45 ka to 7.45 ka. This explanation will be written in the revised manuscript in Chap 3.3.

A sentence added to section 3.3: To calculate correlation coefficients between records, we selected data from 11.45 ka to 7.45 ka.

(c) before interpreting very small variations in CO2 it is important to show they are robust, ie observed at different sites. As rev 2 says this will really only be tested when we have other data, but you can do more with what you have. You already dismiss the WAIS Divide data but you don't actually let the reader see the crucial comparison even for EDC-SD. Thus the reader cannot judge whether your statement "We observe that CO2 data sets from Siple Dome and Dome C share similar trends in CO2 variations" is correct. So please add a figure (I would propose in the main text (not supplement), maybe as another panel to Fig 2) in which you produce the filtered record (as in Fig 1B) for all 3 sites. When you have done this please discuss seriously how robust your findings are, and exercise an appropriate caution in the rest of the paper depending on the result.
Revised

[Figure]

**Figure R3.** A. Atmospheric $CO_2$ records. Red dots: Atmospheric $CO_2$ record from Dome C ice core. Red line: 250-yr running means of atmospheric $CO_2$ record from Dome C ice core. Blue dots: Atmospheric $CO_2$ record from Siple Dome ice core. Blue line: 250-yr running means of atmospheric $CO_2$ record from Siple Dome ice core. Green dots: Atmospheric $CO_2$ record from WAIS Divide ice core. Green line: 250-yr running means of atmospheric $CO_2$ record from WAIS Divide ice core. B. Blue line indicates 250-yr running means of the original Siple Dome $CO_2$ data processed by high-pass filtering at $1/1800$ $yr^{-1}$. Green line indicates 250-yr running means of the original WAIS Divide $CO_2$ data processed by high-pass filtering at $1/1800$ $yr^{-1}$. Red line indicates 250-yr running means of the original WAIS Divide $CO_2$ data processed by high-pass filtering at $1/1800$ $yr^{-1}$. C. $CO_2$ offset between Siple Dome $CO_2$ record and other published $CO_2$ records. Red line: $CO_2$ offset between Siple Dome $CO_2$ record and Dome C $CO_2$ record. Green line: $CO_2$ offset between Siple Dome $CO_2$ record and WAIS divide $CO_2$ record.

(d) Line 164 and line 15 "The Siple Dome CO2 record shows millennial variability of ~2–6 ppm". Looking at Fig 1B, the maximum variation is clearly only 4 ppm, please correct.

Revised

4. The comparison with other records is OK to make (Fig 3) as long as you have caveated about how robust the variations you see are (as per my previous comments). However again please be honest and cautious. While you get reasonable correlations with 14C and 10be, it is nonetheless the case that only 2 of your 3 serious dips have an expression in your solar proxies. At 9.1k, the solar proxies are antiphased with CO2. You should mention this. Taken together with the discussion of later solar variations (around line 250 and discussed by rev 2), these should cause you to caution that the link with solar is very speculative.

Revised

**Section 4 revised to :** In this study, we observed that atmospheric $CO_2$ is highly anti-correlated with the $^{14}C$ production rate and $^{10}Be$ flux with $CO_2$ time lag during the early Holocene (Figure 3). However it is the case that large variations of solar forcing at ~11.1, 10.1 and 8.3 ka. The $^{14}C$ production rate and $^{10}Be$ flux are correlated with $CO_2$ at ~9.1 ka on submillennial time scales.

5. Please redraft section 4 and the abstract to be very cautious based on all the above. In particular the sentence "These relationships suggest that weak solar forcing changes might have impacted CO2 by changing CO2 outgassing from the Southern Ocean and the East Equatorial Pacific and terrestrial carbon storage in the Northern Hemisphere over the early Holocene" suggests you have established a mechanism which is not the case. I am OK with you making the case that there is a tentative correlation with solar forcing but in the abstract you should not go further.

**Section 4 revised to:** Section 4 revised to : In this study, we observed that atmospheric $CO_2$ is highly anti-correlated with the $^{14}C$ production rate and $^{10}Be$ flux with $CO_2$ time lag during the early Holocene (Figure 3). However it is the case that large variations of solar forcing at ~11.1, 10.1 and 8.3 ka. The $^{14}C$ production rate and $^{10}Be$ flux are correlated with $CO_2$ at ~9.1 ka on submillennial time scales.

We also check the correlation of CO2 with solar activity during the last 2,000 years on centennial time. A positive correlation between solar forcing and atmospheric $CO_2$ is observed during the Little Ice Age (LIA). There are two periods in which sunspots were exceedingly rare. During the Maunder sunspot minimum (1647–1715 CE), total solar irradiance (TSI) was reduced by 0.85±0.16 W m$^{-2}$. Atmospheric $CO_2$ records from Antarctic ice cores commonly show a decrease trend during this period (Ahn et al., 2012; Monnin et al., 2004; Siegenthaler et al., 2005; Rubino et al., 2019). During the Spörer Minimum (1450–1550 CE), TSI record during this period also shows a decrease trend. However, atmospheric $CO_2$ decrease is not significant in Law Dome and EPICA Dronning Maud Land (EDML) records (Monnin et al., 2004; Siegenthaler et al., 2005; Rubino et al., 2019), while WAIS divide ice record shows a decrease during this period (Ahn et al., 2012) (Figure S7 in SI). However, atmospheric $CO_2$ decrease drastically at ~1600 CE when total solar irradiance (TSI) shows a local maximum, which is similar to the relationship between solar forcing and atmospheric $CO_2$ at ~9.1 ka. To conclude, it is vague how solar forcing is related with atmospheric $CO_2$ variations on millennial time scales.

However, comparing the early and last Holocene requires attention due to different boundary conditions during these two periods and anthropogenic $CO_2$ during the late Holocene (e.g., Ruddiman, 2003, 2007). Variations of solar forcing are large on a centennial time scale during the Early Holocene. Thus, the solar output effect might be enhanced since the climate system is not responded linearly (Mohtadi et al., 2016). However, due to a decrease in summer insolation and the small variation of solar forcing the late Holocene (7–1 ka) (Berger, 1978), solar forcing might play a less important role during the late Holocene. Further studies are needed to understand the relationship between atmospheric $CO_2$ and solar forcing on shorter time scales during the Early Holocene with more proxy records and numerical models.

**Abstract revised to:**

We present a new high-resolution record of atmospheric $CO_2$ from the Siple Dome ice core, Antarctica over the early Holocene (11.7–7.4 ka) that quantifies natural $CO_2$ variability on millennial timescales under interglacial climate conditions. Atmospheric $CO_2$ decreased by ~10 ppm between 11.3 and 7.3 ka. The decrease was punctuated by local minima at 11.1, 10.1, 9.1 and 8.3 ka with amplitude of 2–4 ppm. Although the linkage between atmospheric $CO_2$ and the climate change remains uncertain due to insufficient paleoclimate records and model simulations, these variations correlate with proxies for solar forcing and local climate in the South East Atlantic polar front, East Equatorial Pacific and North Atlantic. Additional $CO_2$ measurements from a higher accumulation site and carbon cycle models are needed.

**References**

Bouttes, N., Roche, D. M., and Paillard, D.: Systematic study of the impact of fresh water fluxes on the glacial carbon cycle, Clim. Past, 8, 589–607, https://doi.org/10.5194/cp-8-589-2012, 2012.

Marchitto, T. M., Muscheler, R., Ortiz, J. D., Carriquiry, J. D., & van Geen, A. (2010). Dynamical response of the tropical Pacific Ocean to solar forcing during the early Holocene. *Science*, *330*(6009), 1378-1381.

Menviel, L., England, M. H., Meissner, K., Mouchet, A., and Yu, J.: Atlantic-Pacific seesaw and its role in outgassing $CO_2$ during Heinrich events, Paleoceanography, 29, 58–70, 2014.

Mohtadi, M., Prange, M., and Steinke, S.: Palaeoclimatic insights into forcing and response of monsoon rainfall, Nature, 533, 191–199, 2016.

Schmidt, M. W., Weinlein, W. A., Marcantonio, F., and Lynch-Stieglitz, J. (2012), Solar forcing of Florida Straits surface salinity during the early Holocene, *Paleoceanography*, 27, PA3204, doi:10.1029/2012PA002284.

Schmittner, A. and Galbraith, E. D.: Glacial greenhouse-gas fluctuations controlled by ocean circulation changes, Nature, 456, 373, https://doi.org/10.1038/nature07531, 2008.

Yang, J.-W., Ahn, J., Brook, E. J., and Ryu, Y.: Atmospheric methane control mechanisms during the early Holocene, Clim. Past, 13, 1227–1242, https://doi.org/10.5194/cp-13-1227-2017, 2017.

---

## Author Response (AR3)

Dear editor,

We thank the editor for the careful review of our paper, and the suggestions. Our detailed responses to the comments are shown in blue, and the resulting changes to the manuscript are shown in green.

On behalf of all co-authors,

Jinhwa Shin

Comments to the author:

Thank you for your comments on the reviews and for your revisions to the manuscript. Although I share the feeling of the reviewers that you are overstating the robustness of the millennial variations you discuss, i think there are now enough caveats that the data can safely be published and readers can draw their own conclusions. There are two issues where further edits are needed. I refer to line numbers in the clean, final pdf:

1. Line 256-7. "However it is the case that large variations of solar forcing at ~11.1, 10.1 and 8.3 ka. The 14C production rate and 10Be flux are correlated with CO2 at ~9.1 ka on submillennial time scales." This doesn't make sense - the first sentence doesn't finish and the second sentence refers to a correlation at a single time point. Please check and edit this paragraph.

The paragraph is revised to: In this study, we observed that atmospheric $CO_2$ is highly anti-correlated with the [14]C production rate and [10]Be flux on millennial time scales with $CO_2$ time lag during the early Holocene (Figure 3). The local minima of atmospheric $CO_2$ highly match with the local maxima of the [14]C production rate and [10]Be flux (minima in solar activity) at ~11.1, 10.1 and 8.3 ka. The phenomena might be related to large variations in solar activity. However, the relationship between solar forcing and atmospheric $CO_2$ is different at ~9.1 ka. The [14]C production rate and [10]Be flux are positively correlated with $CO_2$ at ~9.1 ka on sub-millennial time scales, indicating that atmospheric $CO_2$ was in a local minimum at ~9.1 ka when solar forcing was relatively high.

2. I understand that you restricted correlations between CO2 and other climate records to 11.45-7.45 ka. Did you also do this for the correlations (lines 115 and 128) between ice core records? In any case, what is relevant for assessing whether the millennial variations you see are robust or not is the correlation of the filtered/detrended records (as shown in Fig 2B). I would be very surprised if these are as high as you cite. Please cite the correlation coefficients of the filtered records. Please also reconsider the phrase (line 126) "We observe that CO2 data sets from Siple Dome and Dome C share similar trends in CO2 variations despite the CO2 offset in longer term means of 3–8 ppm". To me the blue line (SD) and the red line (EDC) do not share the same millennial peaks, rather they are offset, and you should acknowledge that.

Yes, I calculated correlations between Siple Dome and other $CO_2$ records from WAIS Divide and Dome C with their 250-running means. As you suggested I also calculated the correlations with the filtered $CO_2$ records.

Line 116 is revised to: The correlation coefficient between Siple Dome $CO_2$ and WAIS divide $CO_2$ during 11.45–9.02 ka is 0.02 (p =0.28)

Line 126 is revised to: The $CO_2$ record from the Siple Dome is roughly correlated with the $CO_2$ record from Dome C during 11.45–7.45 ka (r= 0.42, p < 0.001). We observe the $CO_2$ offset of 3-8 ppm in the 250-yr running means.

Please address these two points and I should be able to accept the paper.